# On the Robustness of Graph Neural Diffusion to Topology Perturbations

**Yang Song**[*]
Nanyang Technological University
C3 AI
yang.song@c3.ai

**Qiyu Kang**[*]
Nanyang Technological University
kang0080@e.ntu.edu.sg

**Sijie Wang**[*]
Nanyang Technological University
wang1679@e.ntu.edu.sg

**Kai Zhao**[*]
Nanyang Technological University
kai.zhao@ntu.edu.sg

**Wee Peng Tay**
Nanyang Technological University
wptay@ntu.edu.sg

## Abstract

Neural diffusion on graphs is a novel class of graph neural networks that has attracted increasing attention recently. The capability of graph neural partial differential equations (PDEs) in addressing common hurdles of graph neural networks (GNNs), such as the problems of over-smoothing and bottlenecks, has been investigated but not their robustness to adversarial attacks. In this work, we explore the robustness properties of graph neural PDEs. We empirically demonstrate that graph neural PDEs are intrinsically more robust against topology perturbation as compared to other GNNs. We provide insights into this phenomenon by exploiting the stability of the heat semigroup under graph topology perturbations. We discuss various graph diffusion operators and relate them to existing graph neural PDEs. Furthermore, we propose a general graph neural PDE framework based on which a new class of robust GNNs can be defined. We verify that the new model achieves comparable state-of-the-art performance on several benchmark datasets.

## 1 Introduction

Deep learning on graphs and Graph Neural Networks (GNNs), in particular, have achieved remarkable success in a variety of application areas such as those related to chemistry (molecules), finance (trading networks) and the social media (the Facebook friend network) [1–5]. GNNs have been applied to various tasks including node classification [3], link prediction [6], and recommender systems [7]. The key to the success of GNNs is the neural message passing scheme [8] where messages are propagated along edges and optimized toward a downstream task.

While aggregating neighboring nodes' information is a powerful principle of representation learning, the way that GNNs exchange information between nodes makes them vulnerable to adversarial attacks [9]. Adversaries can perturb a graph's topology by adding or removing edges [10–12] or by injecting malicious nodes to the original graph [13–15]. Another common attack is to perturb node attributes [9, 16–18]. Our paper will mainly tackle graph topology perturbation. Adversaries who can

---

[*]Equal contribution.

36th Conference on Neural Information Processing Systems (NeurIPS 2022).

inject nodes to the original graph while not modifying the original graph directly are called injection attacks [13–15]. Adversaries who can directly modify the original graph including edges and node features are called modification attacks [10–12, 9, 16–18]. To defend against adversarial attacks, several robust GNN models have been proposed. Examples include RobustGCN [19], GRAND [20], and ProGNN [21]. In addition, pre-processing based defenders include GNN-SVD [22] and GNNGuard [23].

Recent studies [24–27] have applied neural Ordinary Differential Equations (ODEs) [28] to defend against adversarial attacks. Some works like [24, 27] have revealed interesting intrinsic properties of ODEs that make them more stable than conventional convolutional neural networks (CNNs). Neural Partial Differential Equations (PDEs) have also been applied to graph-structured data [29, 30]. Some papers like [29] (GRAND) and [30] (BLEND) approach deep learning on graphs as a continuous diffusion process and treat GNNs as spatial discretizations of an underlying PDE. However, robustness to adversarial attack has not been studied on such graph neural PDEs. In this work, we demonstrate that graph neural PDEs are intrinsically more robust against adversarial topological perturbations compared to other GNNs. We investigate the heat diffusion on a general Riemannian manifold and show that the diffusion process is essentially stable under small perturbations of the manifold metric. In doing so, we provide insights into the underlining reasons why graph neural PDEs are stable under graph topological perturbations. Such insights indicate that further improvements to the design of graph neural PDEs are possible. We develop several such improved models under a general graph neural PDE framework and show that these models are also robust to node attribute perturbations.

**Main contributions**. In this paper, our objective is to develop a general diffusion framework on graphs and study the robustness properties of the induced graph neural PDEs. Our main contributions are summarized as follows:

- We review the notion of heat diffusion on Riemannian manifolds and the stability of its semigroup. We present analogous concepts of gradient, divergence, and Laplacian operators for heat diffusion on a graph.
- We generalize heat flow to more general flow schemes, including mean curvature flow and Beltrami flow, which are able to preserve inter-class edges in diffusion. A novel class of graph neural PDEs is thereby induced.
- We show that the proposed graph neural PDEs are intrinsically robust to graph topology perturbations. We verify that the new model achieves comparable state-of-the-art performance on several benchmark datasets.

The rest of this paper is organized as follows. In Section 2, we start with preliminaries on continuous diffusion over a Riemannian manifold and the discrete graph. In Section 3, we present our main results on the stability properties of heat flow on graphs and generalize heat flow to edge-preserving flows from which a new class of graph neural PDEs is proposed. Our proposed model architecture is detailed in Section 4. We summarize experimental results in Section 5 and conclude the paper in Section 6. The proofs for all theoretical results in this paper are given in the supplementary material, where more experiments are also presented.

## 2  Preliminaries

Similar to [29, 30], we consider a graph as a discretization of a Riemannian manifold. We introduce concepts and notations for flows diffused over a general manifold and a discrete graph. Readers are referred to [31] for more details. The stability of the heat kernel and heat semigroup of the heat diffusion equation under perturbations of the manifold metric is considered. This discussion sheds light on the diffusion stability of graphs under perturbation of the graph topology in the next section.

### 2.1  Heat Equation and Solution Stability on Manifold

A Riemannian manifold $(M, g)$ is a smooth manifold $M$ endowed with a Riemannian metric $g$, where the norms of tangent vectors and the angles between them are defined by an inner product. It is the natural generalization of Euclidean space and correspondingly the divergence, gradient, and Laplace operators are well-defined, analogous to their corresponding concepts in Euclidean space. Let $C_0^\infty(M)$ denote the space of smooth functions on $M$ with compact support and $\circ$ the composition operation.

**Definition 1.** *The* Laplace operator $\Delta : C_0^\infty(M) \mapsto C_0^\infty(M)$ *on a d-dimensional Riemannian manifold* $(M, g)$ *is defined as*

$$\Delta = \operatorname{div} \circ \nabla,$$

*where* div *is the* divergence *defined for the* $C_0^\infty$ *vector fields on M and* $\nabla$ *is the* gradient *operator. In a local chart U with coordinates* $(x^1, x^2, \ldots, x^n)$, *for function f and vector field* $\psi$, *we have*

$$(\nabla f)^i = \sum_{j=1}^d g^{ij} \frac{\partial f}{\partial x^j}, \quad \operatorname{div} \psi = \sum_{j=1}^d \frac{1}{\sqrt{\det g}} \frac{\partial}{\partial x^j} \left( \sqrt{\det g} \psi^j \right), \tag{1}$$

*where* $\psi^j$ *is the j-th component of* $\psi$, $g^{ij}$ *the components of the inverse metric of g, and* $\det$ *is the determinant operator of a matrix. We also have*

$$\Delta = \sum_{i,j}^d \frac{1}{\sqrt{\det g}} \frac{\partial}{\partial x^i} \left( \sqrt{\det g} g^{ij} \frac{\partial}{\partial x^j} \right). \tag{2}$$

The classical Cauchy problem associated with the heat diffusion equation is to find a function $\varphi(t, x) \in C_0^\infty(\mathbb{R}^+ \times M)$ such that

$$\begin{cases} \frac{\partial \varphi}{\partial t} = \Delta \varphi, \ t > 0, \\ \varphi|_{t=0} = f, \end{cases} \tag{3}$$

where we only consider $f \in C_0^\infty$ and $\mathbb{R}^+$ is the space of positive real numbers.

We can extend the Laplace operator to the generalized *Dirichlet Laplace operator*, denoted as $\mathcal{L} = -\Delta$ and defined on a larger domain[2] [31, section 4.2]. This operator is self-adjoint and non-negative definite on $L^2(M)$, and it can be shown that the above problem (3) is solved by means of the following family $\{P_t\}_{t \geq 0}$ of operators:

$$P_t := e^{-t\mathcal{L}} = \int_{\operatorname{spec}\mathcal{L}} e^{-t\lambda} \, dE_\lambda = \int_0^\infty e^{-t\lambda} \, dE_\lambda, \tag{4}$$

where $\{E_\lambda\}_{\lambda \in \operatorname{spec}\mathcal{L}}$ in $L^2(M)$ is the unique *spectral resolution* of the Dirichlet Laplace operator $\mathcal{L}$ and $\operatorname{spec}\mathcal{L}$ is the *spectrum set* of $\mathcal{L}$. The family $\{P_t\}_{t \geq 0}$ is called the *heat semigroup* associated with $\Delta$, and the solution is given by $P_t f$. It is also well-known that the solution of the above problem has an integration form via the (minimal) heat kernel function $k_t(x, y) : \mathbb{R}^+ \times M \times M \mapsto \mathbb{R}$ [31, Theorem 7.13]:

$$P_t f(x) = \int_M k_t(x, y) f(y) \, d\mu(y),$$

where $d\mu$ is the Riemannian measure [31, Section 3.4] on $M$.

For a Riemannian manifold $M$, different metrics $g$ give rise to different structures. Let $\widetilde{\Delta}$ be the Laplacian associated with another metric $\tilde{g}$ on $M$ such that, for some $\alpha \geq 1$,

$$\alpha^{-1} \tilde{g} \leq g \leq \alpha \tilde{g}.$$

The new $\widetilde{\Delta}$ is said to be *quasi-isometric* [32] to $g$, and can be viewed as a *perturbation* of the metric $g$. $\widetilde{\Delta}$ can be also viewed as a uniformly elliptic operator [32] with respect to (w.r.t.) to $g$. The stability of the heat semigroup and the heat kernel under perturbations of the Laplace operator (i.e., changes of $g$) is well studied in [33]. In the special case of uniformly elliptic operators on Euclidean manifold $\mathbb{R}^d$, the uniformly elliptic operator can be written as $\operatorname{div}(A\nabla)$ with symmetric measurable coefficients $A(x) = (a_{ij}(x))$ such that $\sum_{i,j} a_{ij}(x) \xi_i \xi_j \geq \lambda_0 |\xi|^2$ for all $\xi \in \mathbb{R}^d$ and for some constant $\lambda > 0$ independent of $x$. The reference [33] shows the following theorem:

**Theorem 1.** *On the Euclidean manifold* $\mathbb{R}^n$, *let* $\{P_t\}_{t \geq 0}$ *and* $\{\widetilde{P}_t\}_{t \geq 0}$ *be two diffusion semigroups on* $\mathbb{R}^d$ ($d \geq 2$) *associated with uniformly elliptic operators* $\operatorname{div}(A\nabla)$ *and* $\operatorname{div}(\tilde{A}\nabla)$ *with symmetric measurable coefficients* $A(x) = (a_{ij}(x))$ *and* $\tilde{A}(x) = (\tilde{a}_{ij}(x))$, *respectively. The corresponding heat kernels are denoted by* $p_t(x, y)$ *and* $\tilde{p}_t(x, y)$. *We then have that*

---

[2]$\mathcal{L}$ is defined on the larger Sobolev space $W_0^2(M)$ [31], and is the (negative) extension of $\Delta$. We have $\mathcal{L} = -\Delta$ on $C_0^\infty$.

i. *There is a bounded, piecewise continuous function $F_1(t, z)$ on $\mathbb{R}^+ \times \mathbb{R}^+$ with $\lim_{z \to 0} F_1(t, z) = 0$ for each $t > 0$ and a constant $c > 0$, both of which depend only on $d$ and $\alpha$, such that*

$$|p_t(x, y) - \tilde{p}_t(x, y)| \leq t^{-d/2} \exp\left(-\frac{\|x - y\|_2^2}{ct}\right) F_1\left(t, \|A - \tilde{A}\|_{L^2_{loc}}\right),$$

*for any $(t, x, y) \in \mathbb{R}^+ \times \mathbb{R}^d \times \mathbb{R}^d$, $\|\cdot\|_2$ is the Euclidean norm, and $\|\cdot\|_{L^2_{loc}}$ is the local $L^2$-norm distance defined in [33].*

ii. *Furthermore, we also have an $L^p$-operator norm bound for $P_t - \widetilde{P}_t$ in terms of the local $L^2$-norm distance between $a_{ij}$ and $\tilde{a}_{ij}$: There is a bounded, piecewise continuous function $F_2(t, z)$ on $\mathbb{R}^+ \times \mathbb{R}^+$ with $\lim_{z \to 0} F_2(t, z) = 0$ for each $t > 0$ that depends only on $d$ and $\alpha$, such that for any $p \in [1, \infty]$, we have*

$$\left\|P_t - \tilde{P}_t\right\|_p \leq F_2\left(t, \|A - \tilde{A}\|_{L^2_{loc}}\right).$$

The above theorem shows the pointwise convergence of the heat kernel under perturbations of the matrix $A$. It proves the stability of the solution under small perturbations of the Laplace operator. These results can also be extended to general Riemannian manifolds [34] since every Riemannian manifold can be isometrically embedded into some Euclidean space [35] and the kernel is isometrically invariant [31, theorem 9.12]. We therefore know that if the difference between $\tilde{A}$ and $A$ (or more generally $\tilde{g}$ and $g$) are small, the final solutions to (3) have small difference. An intuitive explanation of the stability of the solution comes from the fact that the heat kernel $k_t(x, y)$ is the transition density function of a Brownian motion on the manifold [36] since Brownian motion can be viewed as a diffusion generated by half of the Laplace operator. This means that $k_t(x, y)$ is a weighted average over all possible paths between $x$ and $y$ at time $t$, which does not change dramatically under small perturbations of $g$. More specifically, for a Brownian motion on a domain $M$, if we perturb the $g$ over a subset $P \subset M$ then only the paths passing through $P$ will be affected.

If we go one step further from the uniformly elliptic operators $\mathrm{div}(A\nabla)$ defined in Theorem 1 by extending $A(x)\varphi$ to $A(x, t, \varphi, \nabla\varphi)$, we get the following general *quasilinear parabolic equation*:

$$\begin{cases} \frac{\partial\varphi}{\partial t} = \mathrm{div}(A(x, t, \varphi, \nabla\varphi)), \ t > 0 \\ \varphi|_{t=0} = f, \end{cases} \tag{5}$$

where the principle part $A(x, t, \varphi, \nabla\varphi)$ is equipped with additional structure conditions as shown in [37]. When $A(x, t, \varphi, \nabla\varphi) = A(x, t)$ is a time-dependent diffusion process, analogous bounds to Theorem 1 are shown in [38]. We leave the analysis of general $A(x, t, \varphi, \nabla\varphi)$ for future work.

## 2.2 Gradient and Divergence Operators on Graphs

We consider a graph as a discretization of a Riemannian manifold. More specifically, consider an undirected graph $\mathcal{G} = (\mathcal{V}, \mathcal{E})$ consisting of a finite set $\mathcal{V}$ of vertices, together with a subset $\mathcal{E} \subset \mathcal{V} \times \mathcal{V}$ of edges. A graph is weighted when it is associated with a function $w : \mathcal{E} \mapsto \mathbb{R}^+$ which is symmetric, i.e., $w([u, v]) = w([v, u])$, for all $[u, v] \in \mathcal{E}$.

Let $\mathcal{H}(\mathcal{V})$ denote the Hilbert space on $\mathcal{V}$ of real-valued functions with the inner product defined as $\langle a, b \rangle_{\mathcal{H}(\mathcal{V})} = \sum_{v \in \mathcal{V}} a(v)b(v)$, for all $a, b \in \mathcal{H}(\mathcal{V})$. Similarly, we define a Hilbert space $\mathcal{H}(\mathcal{E})$ with inner product $\langle c, d \rangle_{\mathcal{H}(\mathcal{E})} = \sum_{[u,v] \in \mathcal{E}} c([u, v])d([u, v])$, for all $c, d \in \mathcal{H}(\mathcal{E})$. We next define the gradient and divergence operators on graphs analogue to the general continuous manifold defined in Definition 1 as follows [39].

**Definition 2.** *Given an undirected graph $\mathcal{G} = (\mathcal{V}, \mathcal{E})$, we define the following:*

i. *The graph gradient is an operator $\nabla : \mathcal{H}(\mathcal{V}) \mapsto \mathcal{H}(\mathcal{E})$ defined by*

$$\nabla\varphi([u, v]) = \sqrt{\frac{w([u, v])}{h(v)}}\varphi(v) - \sqrt{\frac{w([u, v])}{h(u)}}\varphi(u), \forall [u, v] \in \mathcal{E}, \tag{6}$$

*where $h(v) = \sum_{[u,v] \in \mathcal{E}} w([u, v])$ is the degree of node $v$.*

*ii. The graph divergence is an operator* $\mathrm{div} : \mathcal{H}(\mathcal{E}) \mapsto \mathcal{H}(\mathcal{V})$ *defined by*

$$(\mathrm{div}\,\psi)(v) = \sum_{[u,v]\in\mathcal{E}} \sqrt{\frac{w([u,v])}{h(v)}}\left(\psi([v,u]) - \psi([u,v])\right). \tag{7}$$

*iii. The graph Laplacian is an operator* $\Delta : \mathcal{H}(\mathcal{V}) \mapsto \mathcal{H}(\mathcal{V})$ *defined by*

$$\Delta\varphi = -\frac{1}{2}\mathrm{div}(\nabla\varphi). \tag{8}$$

The graph Laplacian defined above is identical to the normalized Laplacian matrix, i.e.,

$$\Delta = \mathbf{D}^{-1/2}(\mathbf{D} - \mathbf{W})\mathbf{D}^{-1/2}, \tag{9}$$

where $\mathbf{D}$ is a diagonal matrix with $\mathbf{D}(v, v) = h(v)$, and $\mathbf{W}$ is the adjacency matrix satisfying $\mathbf{W}(u, v) = w([u, v])$ if $[u, v] \in \mathcal{E}$ and $\mathbf{W}(u, v) = 0$ otherwise. Note that in (8) we have included the negative sign. The analogue of the graph Laplacian $\Delta$ in Section 2.1 is the Dirichlet Laplace operator $\mathcal{L}$. Note the use of the same notation $\Delta$ for graphs. The manifold Laplace operator $\Delta$ and graph Laplacian $\Delta$ will be apparent from the context.

## 3 Neural Diffusion and Stability on Graphs

We now consider neural diffusion on graphs by making use of concepts from Section 2. Various parabolic-type equations on graphs are studied in this section. The solution stability against adversarial attacks is linked to the stability of the solution for the heat diffusion equation under perturbation of the manifold metric, introduced in Section 2.1. The general framework we consider is based on parabolic-type equations on graphs:

$$\frac{\partial\varphi(u,t)}{\partial t} = \mathrm{div}(A(u, t, \varphi, \nabla\varphi)), \ t > 0 \tag{10}$$

with $\varphi(u, 0)$ being the initial node attribute at node $u$ and where $A(u, t, \varphi, \nabla\varphi)$ can take different forms. We next provide some examples.

### 3.1 Continuous Diffusion on Graphs

**Definition 3** (Heat diffusion)**.** *The heat diffusion on graphs is defined by*

$$\frac{\partial\varphi(u,t)}{\partial t} = \frac{1}{2}\mathrm{div}(\nabla\varphi)(u, t). \tag{11}$$

**Definition 4** (GRAND/BLEND [29, 30])**.** *According to [29, eq (1)] and [30, eq (7) and (9)], the GRAND/BLEND flow is defined by*

$$\frac{\partial\varphi(u,t)}{\partial t} = \frac{1}{2}\mathrm{div}(\nabla_t\varphi)(u, t), \tag{12}$$

*where* $\nabla_t$ *is an adaptive Laplace operator (with possible graph rewiring) depending on the evolved node feature* $\varphi(\cdot, t)$.

The gradient operator defined in GRAND/BLEND assumes constant edge weight. In Definition 4, the weight function $w([u, v])$ defined in (6) is incorporated into the gradient definition, so it can be absorbed in the time-dependent term $\nabla_t\varphi$. Note that BLEND degenerates to GRAND [29] when there is no positional encoding. In this paper, for a fair comparison, we do not use positional encoding for all GNNs.

Analogous to [40], we can define mean curvature flow and Beltrami flow as follows.

**Definition 5.** *The mean curvature diffusion on graphs is defined by*

$$\frac{\partial\varphi(u,t)}{\partial t} = \frac{1}{2}\mathrm{div}\left(\frac{\nabla\varphi}{\|\nabla\varphi\|}\right)(u, t), \tag{13}$$

*where* $-\frac{1}{2}\mathrm{div}\left(\frac{\nabla\varphi}{\|\nabla\varphi\|}\right)$ *is a discrete analogue of the mean curvature operator,* $\|\nabla\varphi\| = \langle\nabla\varphi, \nabla\varphi\rangle_{\mathcal{H}(E)}^{1/2}$ *and* $\|\nabla\varphi(u, t)\| = \left(\sum_{[v,u]\in\mathcal{E}}(\nabla\varphi([u, v], t))^2\right)^{1/2}$.

**Definition 6.** *The Beltrami diffusion on graphs is defined by*

$$\frac{\partial \varphi(u,t)}{\partial t} = \frac{1}{2} \frac{1}{\|\nabla \varphi\|} \mathrm{div} \left( \frac{\nabla \varphi}{\|\nabla \varphi\|} \right) (u,t). \tag{14}$$

Intuitively, the term $\|\nabla \varphi(u,t)\|$, which appears in (13) and (14) but not in (11), measures the smoothness of the signals in the neighborhood around vertex $u$ at time $t$. The diffusion using (13) or (14) at vertex $u$ is small when $\|\nabla \varphi(u,t)\|$ is large, i.e., signals are less smooth around vertex $u$. Hence, mean curvature flow and Beltrami flow are able to preserve the non-smooth graph signals. This phenomenon is visualized in Fig. 1, where mean curvature flow and Beltrami flow are capable of preserving inter-class edges whose weights are much larger than those in heat flow.

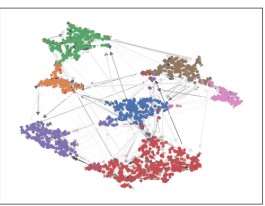 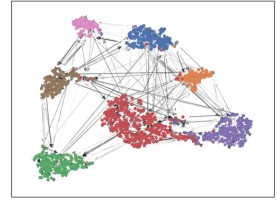 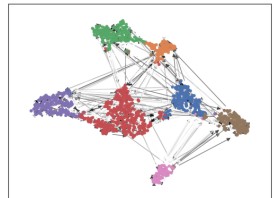

Figure 1: Effect of using different diffusion equations visualized using t-SNE with attention weights. Left: heat flow, middle: mean curvature flow, right: Beltrami flow. The darker the edge, the larger the attention weight.

## 3.2 Stability against Graph Topology Perturbation

Attackers can modify the original graph by adding or removing edges [10–12] and perturbing node attributes. Adding or removing edges leads to a different graph Laplacian $\Delta$ defined in (8). Consequently, the solution to (11) is altered. Using the language of Section 2.1, the edge perturbations correspond to the perturbation of the metric $g$ (or equivalently $A$ in Theorem 1). According to Theorem 1, the stability of the heat semigroup and the solutions under small perturbations of the Laplace operator is guaranteed. Consequently, the solution is not affected significantly by the edge perturbations. This is different from the notion of stability in [20], where the output $\varphi(u,t)$ is shown to be stable after perturbation of node attributes $\varphi(u,0)$ with fixed graph edges. The stability studied in [20] is called *Lyapunov stability*, which is addressed in Proposition 2. These two notions of stability combined lead to a graph neural PDE in Section 4 that is robust against both edge and node attacks.

The following result shows the stability of the solution of (11), i.e., $\frac{\partial \varphi(u,t)}{\partial t} = -\Delta \varphi(u,t)$, under graph topology or Laplacian perturbation. If the change in the graph topology with respect to any matrix norm is small, the semigroup perturbation can be bounded similarly as the result in Theorem 1. This result considers only the case where the Laplace operator is time-invariant. The time-variant analogy of [38] as discussed after (5) is provided in the supplementary material.

**Proposition 1.** *Consider* $\Delta = \mathbf{D}^{-1/2}(\mathbf{D} - \mathbf{W})\mathbf{D}^{-1/2}$ *in* (9) *and suppose* $\tilde{\Delta} = \tilde{\mathbf{D}}^{-1/2}(\tilde{\mathbf{D}} - \tilde{\mathbf{W}})\tilde{\mathbf{D}}^{-1/2}$ *with* $\tilde{\mathbf{D}}$ *being the diagonal degree matrix defined analogously to* (9) *for* $\tilde{\mathbf{W}} = \mathbf{W} + \mathbf{E}$. *Suppose* $\varepsilon := \|\mathbf{E}\| = o(1)$ *for a matrix norm* $\|\cdot\|$, *and* $\mathbf{D}$ *and* $\tilde{\mathbf{D}}$ *are non-singular. Then,* $\|\varphi(u,t) - \tilde{\varphi}(u,t)\| = O(\varepsilon)$.

## 3.3 Neural Flows on Graphs

Substituting (6) and (7) into (14), and ignoring the degree variable $h(u)$ for simplicity, we propose the neural Beltrami flow as

$$\frac{\partial \varphi(u,t)}{\partial t} = \frac{1}{2} \frac{1}{\|\nabla \varphi(u)\|} \sum_{[v,u] \in \mathcal{E}} w([u,v]) \left( \frac{1}{\|\nabla \varphi(u)\|} + \frac{1}{\|\nabla \varphi(v)\|} \right) (\varphi(u) - \varphi(v)), \tag{15}$$

where $\|\nabla \varphi(u)\| = \sqrt{\sum_{[v,u] \in \mathcal{E}} (\nabla \varphi([u,v]))^2}$. Here, we further assume $\varphi(u)$, for all nodes $u$ are time-independent for the sake of simplicity. Suppose $|\mathcal{V}| = n$. The function $\varphi : \mathcal{V} \mapsto \mathbb{R}^d$ maps each

node to a feature vector. Stacking all the feature vectors together, we obtain $\mathbf{Z} \in \mathbb{R}^{n \times d}$. Let the weight function $w([u, v])$ be the scaled dot product attention function [41] given by

$$w([u, v]) = \text{softmax}\left( \frac{(\mathbf{W}_K \mathbf{z}_u)^\intercal (\mathbf{W}_Q \mathbf{z}_v)}{\sqrt{d_K}} \right), \tag{16}$$

where $\mathbf{W}_K$ and $\mathbf{W}_Q$ are the key and query learnable matrices, respectively, and $d_K$ denotes the number of rows $\mathbf{W}_K$ has. If multi-head attention is applied and denote $\mathbf{A}_h$ as the attention matrix associated with head $h$, then $\mathbf{A}(\mathbf{Z}) = \frac{1}{h} \sum_h \mathbf{A}_h(\mathbf{Z})$ with $\mathbf{A}_h(u, v) = w_h([u, v])$ where $w_h$ is the weight function for head $h$. The diffusion equation (15) can be compactly written in matrix form as

$$\frac{\partial \mathbf{Z}(t)}{\partial t} = (\mathbf{A}(\mathbf{Z}(t)) \odot \mathbf{B}(\mathbf{Z}(t)) - \mathbf{\Psi}(\mathbf{Z}(t))) \, \mathbf{Z}(t), \tag{17}$$

where $\odot$ denotes element-wise multiplication, $\mathbf{B}(u, v) = \text{softmax}\left( \frac{1}{\|\nabla \varphi(u)\|^2} + \frac{1}{\|\nabla \varphi(u)\| \|\nabla \varphi(v)\|} \right)$ and $\mathbf{\Psi}(\mathbf{Z}(t))$ is a diagonal matrix with $\Psi(u, u) = \sum_v (\mathbf{A} \odot \mathbf{B})(u, v)$.

Similarly, the diffusion equations using mean curvature flow (13) and heat flow (11) can be written as:

$$\frac{\partial \mathbf{Z}(t)}{\partial t} = (\mathbf{A}(\mathbf{Z}(t)) \odot \mathbf{B}(\mathbf{Z}(t)) - \mathbf{\Psi}(\mathbf{Z}(t))) \, \mathbf{Z}(t), \tag{18}$$

where $\mathbf{B}(u, v) = \text{softmax}\left( \frac{1}{\|\nabla \varphi(u)\|} + \frac{1}{\|\nabla \varphi(v)\|} \right)$ and

$$\frac{\partial \mathbf{Z}(t)}{\partial t} = (\mathbf{A}(\mathbf{Z}(t)) - \mathbf{I})\mathbf{Z}(t), \tag{19}$$

respectively. Although the BLEND model in [30] is inspired from Beltrami flow, its final formulation (equation (8) in [30] where $\mathbf{Z}(t)$ contains positional and node feature embeddings) is indeed (19), i.e., heat flow using attention weight function.

**Proposition 2.** *Diffusion equations* (17)*,* (18) *and* (19) *are all Lyapunov stable [42].*

## 4    Model Architecture with Lipschitz Constraint

In this section, we propose a general graph neural PDE network based on the discussion in Section 3. A layer of graph neural PDE is illustrated in Fig. 2. Since we may stack up multiple such layers, the diffusion is performed in a hierarchical manner, where the high-dimensional features are diffused over the graph at the front layers and the low-dimensional features are diffused at the back layers.

One important ingredient in our model is that we perform spectral normalization on $\mathbf{W}_K$ and $\mathbf{W}_Q$ in (16). This is motivated by the fact that attention models have poor performance when the depth increases. In other words, attention weights tend to be uniformly distributed for excessive message exchanges. This phenomenon becomes even more obvious when the nodes' features are diffused according to (17), (18) or (19) as solving such a PDE normally requires many discrete steps. To overcome this over-smoothing problem, we utilize the strategy proposed in [43] to enforce Lipschitz continuity by normalizing the attention scores. Besides, enforcing Lipschitz continuity can also help improve the robustness of the model because the Lipschitz constant controls the perturbation of the output given a bounded input perturbation. Fig. 3 illustrates that attention weights become overly smooth if no spectral normalization is applied.

Figure 2: Graph neural PDE at the $i$-th layer, where each node's feature vector are linearly transformed before being diffused over the graph. We have $\mathbf{Q} = \mathbf{A}(\mathbf{Z}(t)) \odot \mathbf{B}(\mathbf{Z}(t)) - \mathbf{\Psi}(\mathbf{Z}(t))$ for mean curvature flow and Beltrami flow, while $\mathbf{Q} = \mathbf{A}(\mathbf{Z}(t)) - \mathbf{I}$ for heat flow.

## 5    Experiments

In this section, we compare graph neural PDEs under different flows to popular GNN architectures: GAT [41], GraphSAGE [45], GIN [46], APPNP [47], and the state-of-the-art GNN defenders: Robust-GCN [19], GNNGuard [23], GCNSVD [22], on standard node classification benchmarks. In our experiments,[3] we use the following datasets: Cora (citation networks) [44], Citeseer (citation networks)

---

[3]Our experiments are run on a GeForce RTX 3090 GPU.

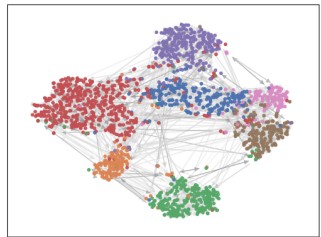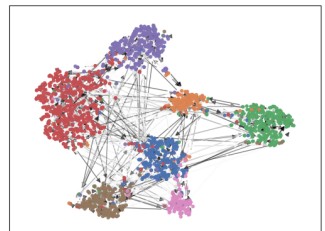

Figure 3: Impact of applying spectral normalization visualized using t-SNE with attention weights using Cora dataset [44]. Left: graph neural PDE with no spectral normalization, right: graph neural PDE with spectral normalization. The darker the edge, the larger the attention weight.

[48] and PubMed (biomedical literature) [49]. We use a refined version of these datasets provided by [50]. We refer the readers to the supplementary material for more details. Our experiment codes are provided in `https://github.com/zknus/Robustness-of-Graph-Neural-Diffusion`.

Table 1: Node classification accuracy (%) on adversarial examples using different GNNs. The implicit Adam PDE solver with step size 2 is used for Beltrami. We denote those experiments that are computationally too heavy to run by "-". The best and the second-best result for each criterion are highlighted in red and blue respectively.

| Dataset | Attack | Beltrami in (17) | RobustGCN | GNNGuard | GCNSVD | GAT | GraphSAGE | GIN | APPNP |
|---|---|---|---|---|---|---|---|---|---|
| Cora | *clean* | 75.93 ± 1.46 | 81.34 ± 0.66 | 79.44 ± 1.18 | 69.28 ± 1.37 | 79.74 ± 1.59 | 76.75 ± 1.52 | 76.79 ± 1.35 | 83.06 ± 1.06 |
| | SPEIT | 61.87 ± 0.49 | 36.16 ± 0.41 | 78.50 ± 2.27 | 37.50 ± 0.74 | 38.10 ± 2.48 | 35.82 ± 0.01 | 35.82 ± 0.01 | 36.79 ± 0.61 |
| | TDGIA | 62.84 ± 1.17 | 53.28 ± 8.61 | 78.92 ± 1.80 | 40.77 ± 3.34 | 35.64 ± 12.91 | 39.78 ± 6.46 | 39.63 ± 2.38 | 60.52 ± 4.43 |
| Citeseer | *clean* | 70.14 ± 1.80 | 70.72 ± 1.15 | 69.69 ± 1.83 | 66.93 ± 1.07 | 69.81 ± 1.43 | 69.78 ± 1.31 | 68.81 ± 1.58 | 70.75 ± 0.86 |
| | SPEIT | 66.46 ± 1.33 | 28.56 ± 7.87 | 69.72 ± 1.84 | 21.16 ± 1.32 | 26.00 ± 11.14 | 23.54 ± 5.30 | 22.19 ± 0.86 |
| | TDGIA | 65.77 ± 1.28 | 38.81 ± 10.84 | 69.50 ± 1.86 | 20.77 ± 2.52 | 19.63 ± 6.53 | 28.77 ± 7.73 | 28.65 ± 5.08 | 54.48 ± 8.56 |
| PubMed | *clean* | 86.94 ± 0.25 | 75.55 ± 0.32 | 84.80 ± 0.51 | - | 84.91 ± 0.76 | 89.22 ± 0.25 | 76.71 ± 0.14 | 77.50 ± 0.54 |
| | SPEIT | 86.66 ± 0.68 | 75.54 ± 0.54 | 84.36 ± 0.58 | - | 40.94 ± 2.47 | 39.22 ± 0.00 | 76.71 ± 0.14 | 77.55 ± 0.54 |
| | TDGIA | 85.56 ± 0.91 | 75.53 ± 0.36 | 84.00 ± 1.12 | - | 39.78 ± 0.29 | 60.40 ± 11.23 | 77.58 ± 0.71 | 77.45 ± 0.68 |

## 5.1 Attack Setup

We apply the setup introduced in the graph robustness benchmark (GRB) [50]. Based on the assumption that nodes with lower degrees are easier to attack, GRB constructs three test subsets of nodes with different degree distributions. According to the average degrees, GRB defines these subsets of nodes as Easy, Medium, Hard, or Full. In the Easy subset, attacks are easy to succeed and hence the worst performance is expected. The remaining nodes are divided into a train set (60%) and val set (10%), for training and validation respectively. In our experiments, we choose the Easy subset, i.e., the most challenging mode.

Following GRB, in this paper, we mainly consider the following real-world adversarial attack settings: *Black-box*: The attacker does not know the defender's method and vice versa. Attackers firstly attack a pre-trained GCN [3] and then transfer the perturbed graphs to the target model. *Evasion*: Attacks will only happen during the inference phase. *Inductive*: GNNs are used to classify unseen data (e.g., new users), i.e., validation or test data are unseen during training. *Injection*: Attackers can only inject new nodes but not modify the target nodes directly. This reflects applications like in online social networks where it is usually hard to hack into users' accounts and modify their profiles. However, it is easier to create fake accounts and connect them to existing users. For other attack settings such as white-box attacks and modification attacks, relevant experiments and discussions are provided in the supplementary material.

Table 2: Node classification accuracy (%) on adversarial examples using BeltramiGuard.

| Model | Attack | Cora | Citeseer | PubMed |
|---|---|---|---|---|
| BeltramiGuard | *clean* | 73.01 ± 2.01 | 69.90 ± 0.44 | 87.77 ± 0.14 |
| | SPEIT | 73.01 ± 2.62 | 69.90 ± 0.44 | 87.68 ± 0.03 |
| | TDGIA | 72.14 ± 1.20 | 69.90 ± 0.44 | 88.16 ± 0.61 |

We apply two state-of-the-art injection attack methods: SPEIT [14] and Topological Defective Graph Injection Attack (TDGIA) [15]. These two attacks are the two strongest attacks reported in the GRB. For both attacks, both the maximum allowable injected nodes and the maximum allowable injected edges are set to 50 for Cora and Citeseer, and 300 for PubMed. We provide more details about how these two attacks work under evasion, black-box, and injection setting in the supplemental material.

Table 1 indicates that graph neural PDE induced from Beltrami flow is more robust than all other GNNs except for GNNGuard, which is specifically designed to remove malicious edges and is thus robust against injection attacks. This suggests that the output features from a graph PDE are stable under topology perturbations, as suggested by Proposition 1.

Furthermore, the heat diffusion process on a manifold tends to diffuse differently under different geometries. For example, it diffuses slower at points with positive curvature, and faster at points with negative curvature [34, 51]. In [52] and [53], the authors show different datasets have different geometric properties like hyperbolicity distribution or Balanced Forman curvature. Our theoretical analysis only shows a loose uniform bound in terms of the adjacency matrix, while the performance is very much dataset-dependent. More advanced theoretical analysis for different datasets is highly non-trivial and needs further investigations.

Our graph neural PDE can be combined with GNNGuard. We denote this model as BeltramiGuard. Table 2 demonstrates that BeltramiGuard renders attacks in vain and exceeds GNNGuard on the Citeseer and PubMed datasets. Note from Table 1 that the vanilla Beltrami model already surpasses GNNGuard on the large PubMed dataset. More details are provided in the supplementary material.

## 5.2 Ablation Studies

### Diffusion Schemes

We compare different diffusion equations: heat flow (19) where $w([u, v]) = 1$ for all $[u, v] \in \mathcal{E}$, GRAND/BLEND [29, 30] which is another heat flow where $w([u, v])$ in (19) is the attention function defined in (16), mean curvature flow (18), and Beltrami flow (17). For GRAND/BLEND, we stack three neural PDE layers using the architecture proposed in Fig. 2 since the original GRAND/BLEND in [29, 30], which has only one PDE layer, does not perform well.

Table 3: Node classification accuracy (%) on adversarial examples using graph neural PDEs induced from different flows, where implicit Adam PDE solver with step size 2 is used.

| Dataset | Attack | Heat | GRAND/BLEND | Mean Curvature | Beltrami |
|---|---|---|---|---|---|
| Cora | *clean* | 78.86 ± 1.78 | 74.89 ± 1.29 | 76.01 ± 2.20 | 75.93 ± 1.46 |
| | SPEIT | 38.66 ± 2.40 | 55.67 ± 3.60 | 60.67 ± 1.31 | 61.87 ± 0.49 |
| | TDGIA | 60.26 ± 5.19 | 59.55 ± 6.41 | 62.01 ± 2.37 | 62.84 ± 1.17 |
| Citeseer | *clean* | 69.47 ± 1.22 | 69.45 ± 0.91 | 70.50 ± 1.63 | 70.14 ± 1.80 |
| | SPEIT | 22.95 ± 5.07 | 39.94 ± 6.94 | 65.39 ± 1.62 | 66.46 ± 1.33 |
| | TDGIA | 52.42 ± 11.2 | 54.92 ± 7.64 | 66.83 ± 1.59 | 65.77 ± 1.28 |
| PubMed | *clean* | 86.31 ± 0.46 | 88.89 ± 0.38 | 88.45 ± 0.32 | 86.94 ± 0.25 |
| | SPEIT | 40.77 ± 1.78 | 39.39 ± 0.26 | 87.13 ± 0.33 | 86.66 ± 0.68 |
| | TDGIA | 42.63 ± 5.28 | 63.91 ± 11.61 | 85.79 ± 0.82 | 85.56 ± 0.91 |

From Table 3, we observe that even the vanilla time-invariant heat flow preserves some robustness as compared to non-PDE GNNs in Table 1. This further validates our theoretical analysis in Proposition 1, which suggests that if the topology perturbation is bounded, the learned representations are close to those under the "clean" scenario. We can observe that 1) the flows which are capable of preserving non-smooth features are generally more robust than heat flow, 2) the proposed mean curvature flow and Beltrami flow are more robust than GRAND/BLEND, and 3) these two flows generally suffer less performance variance than the other methods.

### More Experiments

Due to space constraint, we refer the reader to the supplementary material for more ablation studies including the impact of the Lipschitz constraint, PDE solvers, the number of layers, and time complexity. The model performance under other attacks is also presented in the supplementary material.

# 6    Conclusion

In this paper, we have introduced a general graph neural PDE framework from which several graph neural PDEs are proposed. We analyzed the robustness of the graph neural PDEs and showed that graph neural PDEs are inherently robust against topology perturbations and are also Lyapunov stable. We provided theoretical evidence showing that the robustness of graph neural PDEs stems from the stability of the heat kernel and semigroup during the diffusion process. Moreover, we conducted extensive experiments that empirically verify the robustness of the proposed graph neural PDEs when compared with an existing graph neural PDE induced from approximated heat flow, popular GNNs and the state-of-the-art GNN defenders.

# 7    Acknowledgement

This research is supported by the Singapore Ministry of Education Academic Research Fund Tier 2 grant MOE-T2EP20220-0002 and A*STAR under its RIE2020 Advanced Manufacturing and Engineering (AME) Industry Alignment Fund – Pre Positioning (IAF-PP) (Grant No. A19D6a0053).

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
