# Supplement

**Yang Song**[*]
Nanyang Technological University
C3 AI
`yang.song@c3.ai`

**Qiyu Kang**[*]
Nanyang Technological University
`kang0080@e.ntu.edu.sg`

**Sijie Wang**[*]
Nanyang Technological University
`wang1679@e.ntu.edu.sg`

**Kai Zhao**[*]
Nanyang Technological University
`kai.zhao@ntu.edu.sg`

**Wee Peng Tay**
Nanyang Technological University
`wptay@ntu.edu.sg`

In Section S1 of this supplementary material, we provide more details about the datasets used in our main paper. We briefly describe the attack settings in Section S2. More experiments and ablation studies that are not included in the main paper due to space constraints are now presented in Section S3. We include supplemental experiments with more datasets in Section S4. Further ablation studies of our model are also presented in Section S5. In addition, we present an extension of mean curvature and Beltrami flows in Section S6. Additional implementation details of the models are provided in Section S7. The proofs for all theoretical results in the main paper are given in Section S8.

## S1    Datasets

In our main paper, we conduct experiments using the first three datasets in Table S1: Cora (citation networks) [1], Citeseer (citation networks) [2] and PubMed (biomedical literature) [3]. In this supplementary material, we conduct further experiments using the other three datasets in Table S1: Flickr (social networks) [4], Coauthor (academic networks) [5] and Amazon Computer (recommendation networks) [6]. We use a refined version of these datasets provided by [7], where the main statistics are summarized in Table S1. The features in these datasets are normalized by an $\arctan$ (bijective) transformation [7], which permits attackers to restore the original features and thus allows real-world adversarial attacks.

Table S1: Statistic of Datasets

| Dataset | # Nodes | # Edges | # Features | # Classed | Feature Range (norm) |
|---------|---------|---------|------------|-----------|----------------------|
| Cora | 2,680 | 5,148 | 302 | 7 | $-0.94 \sim 0.94$ |
| Citeseer | 3,191 | 4,172 | 768 | 6 | $-0.96 \sim 0.89$ |
| PubMed | 19,717 | 44,325 | 500 | 3 | $-0.14 \sim 0.99$ |
| Coauthor | 18,333 | 81,894 | 6,805 | 15 | $-0.04 \sim 1.00$ |
| Flickr | 89,250 | 449,878 | 500 | 7 | $-0.47 \sim 1.00$ |
| Amazon Computer | 13,752 | 245,861 | 767 | 10 | $-0.40 \sim 0.60$ |

---

[*]Equal contribution.

36th Conference on Neural Information Processing Systems (NeurIPS 2022).

## S2   Attack Settings

There are three common categories of adversarial attacks studied in the literature [7]:

- Poison (attack occurs in training) or evasion (attack occurs in testing).
- White-box (attackers knows target model/method) or black-box (attackers do not know target model/method and thus need to attack a surrogate model and then transfer to target model).
- Injection (attackers inject nodes/edges to the original graph and generate attributes for the injected nodes) or modification (attackers modify the original graph including its topology and node features directly).

Our experiments in the main paper mainly focus on the evasion, black-box, and injection attack setting, which we believe are the most realistic attack settings. The two selected attack methods SPEIT and TDGIA are tailored for these attack settings. To be more specific, we carry out the following:

- Evasion: SPEIT and TDGIA are performed on a trained model during testing time.
- Black-box: SPEIT and TDGIA are used to attack a trained GCN, i.e., a surrogate model, to generate graph perturbations, and then the target model is tested on this perturbed graph.
- Injection: when perturbing the graph based on a trained GCN, SPEIT and TDGIA first inject new nodes into the original graph and then generate the injected nodes' features.

In this supplementary material, we include more attack settings such as white-box attacks and modification attacks. Further experiments to test graph PDE robustness against node attribute perturbation are also included.

## S3   More Experiments and Ablation Studies

We include more experiments and ablation studies that are not included in the main paper due to space constraints. More specially, in this section, we conduct experiments to study the effect of the Lipschitz constraint and the number of layers, inference time complexity, more injection attacks including evasion white-box injection attacks and attacks with various attack strengths, and the modification attackers which modify the original graph including its topology and node features directly.

Our codes are developed based on the following two repositories:

- `https://github.com/twitter-research/graph-neural-pde` and
- `https://github.com/THUDM/grb`,

where the new diffusion schemes and their induced neural PDEs are developed based on the first repository and we follow the second repository to set up the robustness evaluation benchmark.

### S3.1   PDE Solvers

The impact of PDE solvers on the performance can be observed in Table S2. The adaptive step-size solver Dopri5 performs better than the fixed-step solvers (Implicit/Explicit Adam) at the cost of higher computational complexity. For fixed-step solvers, increasing the step size $\tau$ reduces the variance. For sufficiently large step sizes, the implicit method converges faster than the explicit method.

### S3.2   Lipschitz Constraint and Number of Layers

On one hand, keeping as few layers as possible can mitigate the over-smoothing problem. On the other hand, too few layers result in the underfitting problem, i.e., the test clean accuracy is low. To understand this better, we performed experiments using a different number of layers. The results are summarized in Table S3. The number of layers is a hyperparameter we tune during training.

We have tried using the same number of layers for GRAND/BLEND as in their paper. However, the test clean accuracy is low. This is mainly due to two reasons: 1) we are using the inductive

Table S2: Node classification accuracy (%) using graph neural PDEs induced from Beltrami flow, when different PDE solvers are applied. Experiments are conducted on Citeseer dataset.

| PDE solvers | Param. | Clean | SPEIT | TDGIA |
|---|---|---|---|---|
| Implicit Adam | $\tau = 1$ | $70.59 \pm 2.26$ | $64.64 \pm 2.60$ | $65.62 \pm 0.96$ |
| | $\tau = 2$ | $70.14 \pm 1.80$ | $66.46 \pm 1.33$ | $65.77 \pm 1.28$ |
| | $\tau = 10$ | $70.22 \pm 0.70$ | $64.14 \pm 1.00$ | $65.75 \pm 1.65$ |
| Explicit Adam | $\tau = 1$ | $69.72 \pm 1.14$ | $62.88 \pm 2.70$ | $64.81 \pm 1.62$ |
| | $\tau = 2$ | $69.59 \pm 0.92$ | $65.05 \pm 1.39$ | $65.38 \pm 1.35$ |
| | $\tau = 10$ | $69.91 \pm 0.96$ | $64.76 \pm 1.34$ | $65.52 \pm 3.17$ |
| Dopri5 | - | $70.91 \pm 0.98$ | $66.96 \pm 1.46$ | $67.01 \pm 1.94$ |

Table S3: Node classification accuracy (%) on adversarial examples generated from SPEIT. We apply graph neural PDEs induced from Beltrami flow with or without the Lipschitz constraint. Experiments are conducted on Cora dataset.

| Clean/Robust acc. | # layers used | Beltrami | Beltrami w/o Lips. |
|---|---|---|---|
| Clean | 1 | $63.96 \pm 1.95$ | $63.51 \pm 3.18$ |
| | 2 | $72.46 \pm 0.76$ | $69.93 \pm 1.48$ |
| | 3 | $73.51 \pm 1.49$ | $72.69 \pm 1.89$ |
| | 4 | $75.93 \pm 1.46$ | $75.15 \pm 1.41$ |
| Robust | 1 | $59.10 \pm 3.20$ | $58.06 \pm 1.78$ |
| | 2 | $63.58 \pm 1.39$ | $58.96 \pm 2.82$ |
| | 3 | $60.52 \pm 1.56$ | $57.46 \pm 1.65$ |
| | 4 | $61.87 \pm 0.49$ | $56.79 \pm 1.52$ |

setting whereas the paper of GRAND/BLEND uses transductive training; 2) the datasets we are using have been calibrated for the robustness evaluation, which is different from the original datasets used by the paper of GRAND/BLEND. For example, grb-cora [7] has node feature size of 302 while the original Cora dataset has node feature size of 1433. Table S3 provides empirical evidence for GRAND/BLEND needing more layers in our setting.

### S3.3 Time Complexity

We have summarized the time complexity in Table S4. The time is computed by averaging 500 diffusion operations. We can see that Beltrami and GRAND/BLEND have similar time complexity to the two defenders GNNGuard and GCNSVD small step sizes but incur more computation time than the other GNNs to complete a diffusion process. Heat, unlike Beltrami and GRAND/BLEND, does not use attention and has the lowest time complexity among all the neural PDEs.

### S3.4 White-Box Attacks

We now include the white-box attacks for our model and the baselines. The results are shown in Table S5. We observe that our model outperforms the baselines under white-box attacks. The results further validate that neural PDEs are intrinsically robust to graph topology perturbations, as indicated by Proposition 1 and Lemma S1.

### S3.5 Injection Attacks with Various Attack Strengths

Under injection attacks with a different number of injection nodes and edges, we have performed more experiments with various attack strengths. Table S6 shows that the robustness is not significantly affected by the number of nodes and edges injected by attackers as the original is not perturbed in this setting.

Table S4: Top: Average time spent on a Beltrami diffusion process, i.e., time to solve (17) and a counterpart in GRAND/BLEND, when different PDE solvers are applied and multiple step size options are tested for each solver. Bottom: Average time spent on an aggregation step using different GNNs. Experiments are conducted on the Citeseer dataset.

| PDE solvers | Param. | Beltrami | GRAND/BLEND | Heat |
|---|---|---|---|---|
| Implicit Adam | $\tau = 1$ | 9.8ms | 6.6ms | 3.0ms |
| | $\tau = 2$ | 17.0ms | 11.2ms | 4.0ms |
| | $\tau = 10$ | 48.1ms | 46.6ms | 9.2ms |
| Explicit Adam | $\tau = 1$ | 10.0ms | 6.8ms | 3.0ms |
| | $\tau = 2$ | 16.8ms | 11.2ms | 3.6ms |
| | $\tau = 10$ | 32.6ms | 21.0ms | 5.8ms |
| Dopri5 | - | 66.0ms | 20.0ms | 13.0ms |

| RobustGCN | GNNGuard | GCNSVD | GAT | GraphSAGE | GIN | APPNP |
|---|---|---|---|---|---|---|
| 0.6ms | 13.2ms | 9.0ms | 1.8ms | 0.8ms | 1.0ms | 1.6ms |

Table S5: Node classification accuracy (%) on adversarial examples generated from *white-box* attacks. The best and the second-best result for each criterion are highlighted in red and blue, respectively.

| Dataset | Attack | BeltramiGuard | Beltrami | RobustGCN | GNNGuard | GCNSVD | GAT | GraphSAGE | GIN | APPNP |
|---|---|---|---|---|---|---|---|---|---|---|
| Cora | *clean* | 73.01 ± 2.01 | 75.93 ± 1.46 | 81.34 ± 0.66 | 79.44 ± 1.18 | 69.28 ± 1.37 | 79.74 ± 1.59 | 76.75 ± 1.52 | 76.79 ± 1.35 | 83.06 ± 1.06 |
| | SPEIT | 71.94 ± 0.31 | 48.95 ± 3.32 | 36.12 ± 0.31 | 80.22 ± 0.91 | 33.06 ± 7.86 | 19.18 ± 9.47 | 17.16 ± 11.30 | 21.49 ± 13.08 | 19.77 ± 10.39 |
| | TDGIA | 67.39 ± 1.97 | 52.61 ± 4.20 | 36.27 ± 0.61 | 78.43 ± 1.10 | 11.94 ± 0.0 | 7.99 ± 4.40 | 21.57 ± 4.82 | 38.06 ± 4.69 | 51.94 ± 5.32 |
| Citeseer | *clean* | 69.90 ± 0.44 | 70.41 ± 1.38 | 70.72 ± 1.15 | 69.69 ± 1.83 | 66.93 ± 1.07 | 69.81 ± 1.43 | 69.78 ± 1.31 | 68.81 ± 1.58 | 70.75 ± 0.86 |
| | SPEIT | 68.78 ± 0.82 | 55.24 ± 6.90 | 20.19 ± 2.61 | 69.22 ± 1.90 | 19.31 ± 3.55 | 14.67 ± 5.05 | 19.81 ± 3.06 | 12.54 ± 6.33 | 20.75 ± 2.32 |
| | TDGIA | 67.96 ± 1.22 | 53.61 ± 14.01 | 18.68 ± 4.06 | 68.40 ± 1.44 | 16.93 ± 2.41 | 14.23 ± 5.05 | 20.69 ± 5.64 | 18.87 ± 3.61 | 25.70 ± 6.78 |
| PubMed | *clean* | 87.77 ± 0.14 | 86.94 ± 0.25 | 75.55 ± 0.32 | 84.80 ± 0.51 | - | 84.91 ± 0.76 | 89.22 ± 0.25 | 76.71 ± 0.14 | 77.50 ± 0.54 |
| | SPEIT | 85.26 ± 1.42 | 85.13 ± 0.79 | 75.07 ± 0.30 | 84.30 ± 1.34 | - | 39.22 ± 0.0 | 40.46 ± 1.69 | 75.58 ± 1.03 | 77.62 ± 0.10 |
| | TDGIA | 81.36 ± 3.09 | 84.88 ± 0.46 | 75.78 ± 0.32 | 83.33 ± 2.91 | - | 37.96 ± 1.82 | 44.85 ± 3.06 | 75.72 ± 0.70 | 77.32 ± 0.44 |

Table S6: Node classification accuracy (%) on adversarial examples generated from SPETI and TDGIA under black-box injection setting where a different number of injected nodes and edges are applied.

| Dataset | Attack | # nods/edges injected | Beltrami | RobustGCN | GNNGuard | GCNSVD | GAT | GraphSAGE | GIN | APPNP |
|---|---|---|---|---|---|---|---|---|---|---|
| Cora | SPEIT | 50/50 | 61.87 ± 0.49 | 36.16 ± 0.41 | 78.50 ± 2.27 | 37.50 ± 0.74 | 38.10 ± 2.48 | 35.82 ± 0.01 | 35.82 ± 0.01 | 36.79 ± 0.61 |
| | SPEIT | 100/100 | 57.18 ± 1.68 | 35.82 ± 0.00 | 79.39 ± 1.41 | 35.82 ± 0.00 | 37.22 ± 1.87 | 35.82 ± 0.00 | 35.82 ± 0.00 | 35.82 ± 0.00 |
| | SPEIT | 150/150 | 55.88 ± 0.56 | 35.82 ± 0.00 | 80.78 ± 0.71 | 35.82 ± 0.00 | 26.12 ± 16.29 | 35.82 ± 0.00 | 35.82 ± 0.00 | 29.85 ± 11.94 |
| | SPEIT | 200/200 | 58.21 ± 2.81 | 35.82 ± 0.00 | 79.38 ± 0.64 | 29.88 ± 11.96 | 29.85 ± 11.94 | 35.82 ± 0.00 | 35.82 ± 0.00 | 22.30 ± 15.83 |
| | TDGIA | 50/50 | 62.84 ± 1.17 | 53.28 ± 8.61 | 78.92 ± 1.80 | 40.77 ± 3.34 | 35.64 ± 12.91 | 39.78 ± 6.46 | 39.63 ± 2.38 | 60.52 ± 4.43 |
| | TDGIA | 100/100 | 62.44 ± 1.56 | 54.11 ± 3.18 | 77.99 ± 2.24 | 42.29 ± 0.43 | 36.94 ± 16.96 | 34.24 ± 9.99 | 37.03 ± 1.59 | 62.50 ± 3.47 |
| | TDGIA | 150/150 | 64.18 ± 0.87 | 52.43 ± 10.84 | 79.39 ± 1.74 | 11.94 ± 0.00 | 39.74 ± 16.09 | 38.34 ± 1.48 | 35.82 ± 0.00 | 61.37 ± 7.07 |
| | TDGIA | 200/200 | 61.48 ± 1.68 | 48.88 ± 6.60 | 80.78 ± 1.89 | 11.94 ± 0.00 | 46.74 ± 12.70 | 38.34 ± 2.50 | 36.19 ± 0.75 | 61.10 ± 0.98 |
| Citeseer | SPEIT | 50/50 | 65.52 ± 2.26 | 28.56 ± 7.87 | 69.72 ± 1.84 | 21.16 ± 1.32 | 26.00 ± 11.14 | 19.75 ± 1.82 | 23.54 ± 5.30 | 22.19 ± 0.86 |
| | SPEIT | 100/100 | 64.74 ± 1.27 | 17.01 ± 6.05 | 69.51 ± 1.61 | 19.75 ± 2.56 | 19.12 ± 1.47 | 18.81 ± 2.74 | 20.61 ± 2.39 | 20.53 ± 1.57 |
| | SPEIT | 150/150 | 65.13 ± 2.27 | 20.69 ± 1.49 | 70.22 ± 1.63 | 18.18 ± 1.81 | 18.97 ± 1.57 | 18.34 ± 1.63 | 19.75 ± 0.00 | 19.75 ± 0.00 |
| | SPEIT | 200/200 | 64.11 ± 0.83 | 15.52 ± 7.19 | 69.36 ± 1.29 | 18.96 ± 3.00 | 19.12 ± 1.26 | 22.10 ± 5.11 | 21.16 ± 4.97 | 17.71 ± 1.37 |
| | TDGIA | 50/50 | 65.77 ± 1.28 | 38.81 ± 10.84 | 69.50 ± 1.86 | 20.77 ± 2.52 | 19.63 ± 6.53 | 28.77 ± 7.73 | 28.65 ± 5.08 | 54.48 ± 8.56 |
| | TDGIA | 100/100 | 65.60 ± 1.68 | 45.61 ± 6.75 | 66.22 ± 5.44 | 25.39 ± 6.33 | 25.08 ± 10.21 | 35.81 ± 11.99 | 29.94 ± 7.26 | 52.74 ± 6.48 |
| | TDGIA | 150/150 | 66.22 ± 2.31 | 39.66 ± 15.80 | 69.36 ± 0.90 | 24.14 ± 3.21 | 19.36 ± 2.14 | 26.57 ± 9.28 | 27.51 ± 6.69 | 59.01 ± 3.93 |
| | TDGIA | 200/200 | 64.66 ± 2.15 | 38.48 ± 9.63 | 69.91 ± 1.54 | 26.49 ± 4.14 | 22.33 ± 11.29 | 26.72 ± 13.41 | 26.73 ± 7.47 | 51.72 ± 5.06 |
| PubMed | SPEIT | 300/300 | 86.66 ± 0.68 | 75.54 ± 0.54 | 84.36 ± 0.58 | - | 40.94 ± 2.47 | 39.22 ± 0.00 | 76.71 ± 0.14 | 77.55 ± 0.54 |
| | SPEIT | 600/600 | 86.80 ± 0.98 | 75.71 ± 0.60 | 86.47 ± 0.58 | - | 34.8 ± 10.99 | 41.10 ± 1.63 | 76.36 ± 0.65 | 77.22 ± 0.43 |
| | SPEIT | 900/900 | 87.10 ± 0.38 | 76.00 ± 0.35 | 86.73 ± 0.48 | - | 39.99 ± 1.55 | 40.78 ± 1.77 | 76.73 ± 0.36 | 77.14 ± 0.70 |
| | SPEIT | 1200/1200 | 87.51 ± 0.58 | 74.75 ± 0.90 | 86.19 ± 0.40 | - | 40.82 ± 1.85 | 40.08 ± 1.63 | 76.70 ± 0.72 | 77.17 ± 0.32 |
| | TDGIA | 300/300 | 85.56 ± 0.91 | 75.53 ± 0.36 | 84.00 ± 1.12 | - | 39.78 ± 0.29 | 60.40 ± 11.23 | 77.58 ± 0.71 | 77.45 ± 0.68 |
| | TDGIA | 600/600 | 86.33 ± 0.84 | 75.84 ± 0.23 | 84.21 ± 0.33 | - | 56.00 ± 9.13 | 70.24 ± 20.07 | 76.38 ± 0.75 | 77.37 ± 0.31 |
| | TDGIA | 900/900 | 87.00 ± 1.47 | 75.91 ± 0.55 | 83.87 ± 1.01 | - | 41.44 ± 10.61 | 52.40 ± 6.03 | 75.56 ± 0.58 | 76.88 ± 0.68 |
| | TDGIA | 1200/1200 | 86.87 ± 1.41 | 75.62 ± 0.82 | 84.35 ± 2.24 | - | 57.16 ± 10.92 | 71.72 ± 14.02 | 76.49 ± 0.80 | 77.23 ± 0.46 |

## S3.6 Modification Attacks

In the modification attack, attackers can directly flip the original graph's edges and perturb the features of the nodes. We apply the PGD method to randomly flip edges and then perturb node features. In Table S7, we observe that

1) the robustness performance starts to break down when the 60% of nodes have their features perturbed by $\epsilon = 0.1$ (the value of features is in [-1,1]) and 60% of edges are flipped, and

2) as long as feature perturbation is small, the robustness can still be retained even if 80% of nodes have their features perturbed and 80% of edges are flipped.

In summary, from those extensive experiment results, we observe that our models are more robust against topology perturbation than feature perturbation.

Table S7: Node classification accuracy (%) on adversarial examples generated from PGD under black-box *modification* setting where a different number of modified nodes and edges are applied. Experiments are conducted on Cora dataset.

| Ratio of nodes/edges modified | Feature perturbation | Beltrami |
|---|---|---|
| 20%/20% | $\epsilon = 0.01$ | $73.73 \pm 0.86$ |
| 40%/40% | $\epsilon = 0.01$ | $73.28 \pm 1.36$ |
| 60%/60% | $\epsilon = 0.01$ | $72.46 \pm 1.48$ |
| 80%/80% | $\epsilon = 0.01$ | $72.31 \pm 2.46$ |
| 20%/20% | $\epsilon = 0.1$ | $61.56 \pm 1.06$ |
| 40%/40% | $\epsilon = 0.1$ | $52.15 \pm 2.31$ |
| 60%/60% | $\epsilon = 0.1$ | $44.03 \pm 0.79$ |
| 80%/80% | $\epsilon = 0.1$ | $40.30 \pm 0.91$ |
| 80%/80% | $\epsilon = 1$ | $39.55 \pm 3.46$ |
| 80%/80% | $\epsilon = 2$ | $37.09 \pm 3.02$ |
| 80%/80% | $\epsilon = 5$ | $37.16 \pm 1.88$ |
| 80%/80% | $\epsilon = 10$ | $37.16 \pm 0.33$ |

## S3.7 More Discussion about Neural Heat Diffusion

"Heat" in Table 3 of the main paper serves as a baseline. Recall that both "Heat" and GRAND/BLEND are derived from heat flow, where GRAND/BLEND uses the attention function to weigh edges while "Heat" uses the constant function, which is not learned from the data and hence is not expected to perform well. In a further experiment, we now treat the constant in "Heat" as a trainable variable that is shared by all edges. We denote this new variant as Heat$^+$ and compare it with GAT and APPNP (which were more robust than "Heat") under SPEIT attack in Table S8. We now see that Heat$^+$ is more robust. This constant variable controls the diffusivity on the graph, analogous to the thermal diffusivity on a manifold. It is interesting to observe such a phenomenon since the experiment indicates that heat diffusivity also affects robustness. We believe further investigations in this direction can be performed in future work.

Table S8: Node classification accuracy (%) on adversarial examples generated from SPEIT.

| Dataset | GNN | Clean | 10/10 | 15/15 | 20/20 | 25/25 | 30/30 | 35/35 | 40/40 | 45/45 | 50/50 |
|---|---|---|---|---|---|---|---|---|---|---|---|
| | Heat$^+$ | $69.78 \pm 0.95$ | $69.09 \pm 1.61$ | $67.77 \pm 1.61$ | $66.27 \pm 2.41$ | $66.27 \pm 2.81$ | $64.26 \pm 1.27$ | $61.88 \pm 3.01$ | $57.30 \pm 4.07$ | $54.61 \pm 5.25$ | $54.61 \pm 2.92$ |
| Citeseer | GAT | $69.81 \pm 1.43$ | $60.56 \pm 4.77$ | $62.01 \pm 4.89$ | $45.52 \pm 10.44$ | $30.34 \pm 5.40$ | $31.16 \pm 6.16$ | $31.03 \pm 12.60$ | $36.11 \pm 13.85$ | $23.13 \pm 3.25$ | $26.00 \pm 11.14$ |
| | APPNP | $70.75 \pm 0.86$ | $68.65 \pm 0.91$ | $66.52 \pm 2.90$ | $59.87 \pm 3.49$ | $56.11 \pm 9.70$ | $56.05 \pm 2.37$ | $51.22 \pm 8.88$ | $38.50 \pm 8.75$ | $31.60 \pm 9.80$ | $22.19 \pm 0.86$ |

# S4 Further Experiments

We repeat the experiments in Table 1 in the main paper but using the three new datasets. The inherent robustness of the proposed graph neural PDEs is again verified by Table S9, especially on the Flickr and Coauthor datasets.

Table S9: Node classification accuracy (%) on adversarial examples generated by the SPEIT method. We denote those experiments that are computationally too heavy to run by "-". The best and the second-best result for each criterion are highlighted in red and blue, respectively.

| Dataset | Attack | Beltrami | RobustGCN | GNNGuard | GCNSVD | GAT | GraphSAGE | GIN | APPNP |
|---|---|---|---|---|---|---|---|---|---|
| Flickr | clean | 49.40 ± 0.12 | 47.66 ± 0.00 | - | - | 54.45 ± 0.57 | 53.50 ± 0.02 | 53.57 ± 0.29 | 54.08 ± 0.14 |
| | SPEIT | 49.79 ± 0.68 | 6.57 ± 0.00 | - | - | 6.57 ± 0.00 | 49.71 ± 0.00 | 49.71 ± 0.00 | 6.57 ± 0.00 |
| | TDGIA | 49.47 ± 0.50 | 52.95 ± 1.58 | - | - | 50.38 ± 0.20 | 50.21 ± 0.11 | 50.14 ± 0.25 | 54.28 ± 0.59 |
| Coauthor | clean | 95.83 ± 0.30 | 87.75 ± 0.23 | 92.56 ± 0.16 | - | 92.75 ± 0.15 | 94.53 ± 0.21 | 84.91 ± 0.32 | 87.67 ± 0.16 |
| | SPEIT | 94.83 ± 0.12 | 87.62 ± 0.29 | 92.56 ± 0.16 | - | 2.59 ± 1.46 | 39.44 ± 13.97 | 39.44 ± 13.97 | 87.66 ± 0.16 |
| | TDGIA | 95.06 ± 0.21 | 87.3 ± 0.29 | - | - | 65.32 ± 13.04 | 87.97 ± 3.72 | 85.12 ± 0.33 | 87.54 ± 0.13 |
| Amazon Computer | clean | 87.86 ± 0.30 | 86.22 ± 0.54 | 88.77 ± 0.05 | 74.79 ± 0.68 | 89.21 ± 0.60 | 89.92 ± 0.33 | 86.44 ± 0.23 | 82.66 ± 1.54 |
| | SPEIT | 84.90 ± 0.61 | 86.33 ± 0.62 | 88.62 ± 0.05 | 26.79 ± 1.25 | 26.88 ± 16.68 | 29.19 ± 9.35 | 86.44 ± 0.23 | 82.62 ± 1.55 |
| | TDGIA | 85.50 ± 0.15 | 86.69 ± 0.51 | - | - | 55.45 ± 23.07 | 63.14 ± 10.59 | 86.65 ± 0.57 | 83.44 ± 1.59 |

## S5 Further Ablation Studies

We repeat the experiments in Table 3 of the main paper but using three new datasets. Similar to what we have stated in the main paper, we observe the advantage of the proposed mean curvature flow and Beltrami flow in terms of robustness against adversarial attacks.

Table S10: Node classification accuracy (%) on adversarial examples using graph neural PDEs induced from different flows, where implicit Adam PDE solver with step size 2 is used.

| Dataset | Attack | Beltrami | Mean Curvature | GRAND/BLEND | Heat |
|---|---|---|---|---|---|
| Coauthor | clean | 95.81 ± 0.38 | 95.66 ± 0.18 | 94.35 ± 0.33 | 92.87 ± 0.55 |
| | SPEIT | 95.10 ± 0.35 | 95.41 ± 0.28 | 66.63 ± 9.76 | 34.18 ± 8.85 |
| | TDGIA | 95.06 ± 0.21 | 95.41 ± 0.17 | 85.97 ± 4.50 | 62.39 ± 15.96 |
| Amazon Computer | clean | 87.86 ± 0.30 | 87.59 ± 0.44 | 90.59 ± 0.35 | 90.59 ± 0.57 |
| | SPEIT | 84.90 ± 0.61 | 84.55 ± 1.03 | 75.91 ± 12.22 | 46.82 ± 14.42 |
| | TDGIA | 85.50 ± 0.15 | 85.34 ± 0.78 | 86.31 ± 2.50 | 76.60 ± 10.48 |
| Flickr | clean | 49.42 ± 0.12 | 47.66 ± 1.51 | 48.52 ± 0.19 | 46.65 ± 1.30 |
| | SPEIT | 49.76 ± 0.63 | 49.74 ± 1.18 | 49.75 ± 0.06 | 49.70 ± 0.02 |
| | TDGIA | 49.47 ± 0.50 | 47.39 ± 1.43 | 48.63 ± 0.09 | 47.05 ± 1.20 |

## S6 Extension

The graph Laplacian and curvature can be generalized to an operator that can be thought of as the discrete analog of the $p$-Laplacian in the continuous case [8, 9]:

$$\frac{\partial \varphi(u,t)}{\partial t} = \frac{1}{2}\mathrm{div}\left(\|\nabla \varphi\|^{p-2}\nabla \varphi\right)(u,t), \tag{S1}$$

where $p = 1, 2$ correspond to the mean curvature and heat (GRAND/BLEND) equations, respectively. Here, we consider the cases where $p > 2$. Note that when $p > 2$, as opposed to the cases where $p \leq 1$, e.g., $p = 1$ for mean curvature and Beltrami flows, the edges that connect nodes with similar features are potentially preserved in the diffusion process. Table S11 shows that the PDE's robustness decreases as $p$ increases.

Table S11: Node classification accuracy (%) on adversarial examples using graph neural PDEs induced from $p$-Laplacian flow where $p = 3, 4$, where implicit Adam PDE solver with step size 2 is used. All experiments are done on Citeseer dataset.

| Attack | $p$-Laplacian ($p = 3$) | $p$-Laplacian ($p = 4$) | Mean Curvature | Beltrami |
|---|---|---|---|---|
| clean | 69.15 ± 1.37 | 67.27 ± 1.14 | 70.50 ± 1.63 | 70.41 ± 1.38 |
| SPEIT | 63.76 ± 0.85 | 62.44 ± 1.14 | 65.39 ± 1.62 | 65.52 ± 2.26 |
| TDGIA | 64.89 ± 0.77 | 63.39 ± 2.00 | 66.83 ± 1.59 | 65.77 ± 1.28 |

## S7 Implementation Details

By default, all the models are implemented using three layers, with layer normalization, 50% dropout at the end of each layer, and hidden feature dimensions $64 - 64$. Here are some notes that should be taken:

- When dealing with the Cora dataset, PDEs induced by Beltrami and mean curvature flows tend to underfit. Thus, we implement them using four layers with hidden feature dimensions $128 - 128 - 128$.

- For all neural PDEs, node features are diffused independently over layers, i.e., each layer solves its own PDE. At each layer, once a diffusion process is over, the parameters in the associated PDE are reset. This operation is effective to alleviate the overfitting problem.

- By default, the integral period for all PDEs is set to $[0, 1]$.

All experiments are repeated 5 to 10 times with different random seeds.

For all the attacks, the maximum number of nodes and edges that are allowed to be perturbed are summarized in Table S12.

Table S12: Statistic of attacks' budgets on each dataset

| Dataset | max # Nodes | max # Edges |
|---|---|---|
| Cora | 50 | 50 |
| Citeseer | 50 | 50 |
| PubMed | 300 | 300 |
| Coauthor | 150 | 300 |
| Amazon Computer | 100 | 200 |
| Flickr | 1000 | 5000 |

GNNGuard implemented in Table 1 of the main paper is based on GCN, where each layer in GNNGuard contains two operations: 1) adjusting attention coefficients by pruning likely fake edges and assigning less weight to suspicious edges; and 2) a normal GCN layer. Regarding BeltramiGuard implemented in Table 2 of the main paper, we replace the second component as mentioned above, i.e., a GCN layer, with a PDE layer induced by Beltrami flow.

## S8 Proofs of Results

In this section, we provide detailed proof of the results stated in the main paper. The time-variant analogy of [10] as discussed after (5) in the main paper is also presented.

### S8.1 Proof of Proposition 1

For the system $\frac{\partial \varphi(u,t)}{\partial t} = -\Delta \varphi(u, t)$, the solution is given by $\varphi(u, t) = e^{-t\Delta} \varphi(u, 0)$. We therefore need to derive a bound for $\|e^{-t\Delta} - e^{-t\tilde{\Delta}}\|$. Note for matrix exponent, in general $e^{X+Y} \neq e^X e^Y$ unless $X$ and $Y$ commute (i.e. $XY = YX$). However, from [11, eq (3.5)], we have

$$\left\| e^{-t\Delta} - e^{-t\tilde{\Delta}} \right\| \leq t \left\| \Delta - \tilde{\Delta} \right\| \left\| e^{-t\Delta} \right\| e^{\|t(\Delta - \tilde{\Delta})\|},$$

where the positive definiteness of $\Delta$ is used. We further analyze $\Delta - \tilde{\Delta}$. Since $\tilde{\mathbf{W}} = \mathbf{W} + \mathbf{E}$ with $\|\mathbf{E}\| = \varepsilon$, we have $\|\mathbf{D} - \tilde{\mathbf{D}}\| = O(\varepsilon)$ because norms for a finite dimensional space are equivalent [12, Theorem 5.4.4.]. Let $\mathbf{E}' = \tilde{\mathbf{D}}^{-1/2} - \mathbf{D}^{-1/2}$. We also have $\|\mathbf{E}'\| = O(\varepsilon)$.

It follows that

$$\left\|\mathbf{\Delta} - \tilde{\mathbf{\Delta}}\right\| = \left\|\mathbf{D}^{-1/2}\mathbf{W}\mathbf{D}^{-1/2} - \tilde{\mathbf{D}}^{-1/2}\tilde{\mathbf{W}}\tilde{\mathbf{D}}^{-1/2}\right\|$$

$$= \left\|\mathbf{D}^{-1/2}\mathbf{W}\mathbf{D}^{-1/2} - (\mathbf{D}^{-1/2} + \mathbf{E}')(\mathbf{W} + \mathbf{E})(\mathbf{D}^{-1/2} + \mathbf{E}')\right\|$$

$$= \left\|\mathbf{E}'(\mathbf{W} + \mathbf{E}')(\mathbf{D}^{-1/2} + \mathbf{E}') + \mathbf{D}^{-1/2}\mathbf{E}(\mathbf{D}^{-1/2} + \mathbf{E}') + \mathbf{D}^{-1/2}\mathbf{W}\mathbf{E}'\right\|$$

$$= O(\varepsilon).$$

We therefore obtain the conclusion that $\left\|e^{-t\mathbf{\Delta}} - e^{-t\tilde{\mathbf{\Delta}}}\right\| = O(\varepsilon t e^{-\rho t})$ for some constant $\rho > 0$ since $\left\|e^{-t\mathbf{\Delta}}\right\| = O(e^{-t\rho})$ according to [11] and the fact that $\mathbf{\Delta}$ is positive definite according to our assumption. The conclusion $\|\varphi(u, t) - \tilde{\varphi}(u, t)\| = O(\varepsilon t e^{-\rho t})$ then follows.

## S8.2 Proof of Proposition 2

*Proof.* Since $\mathbf{A}$ is right stochastic, all eigenvalues of $\mathbf{A} - \mathbf{I}$ have non-positive real parts. By the Lyapunov Stability Theorem [13], the stability of (19) is ensured. Since $\mathbf{\Psi}^{-1}\mathbf{A} \odot \mathbf{B}$ is right stochastic, all eigenvalues of $\mathbf{\Psi}^{-1}\mathbf{A} \odot \mathbf{B} - \mathbf{I}$ have non-positive parts and thus (18) and (17) are stable. $\square$

## S8.3 The Time-Variant Case

In this section, we provide an analysis of the time-variant system as alluded to before Proposition 1. In the time-variant system, we have

$$\frac{\partial \varphi(u, t)}{\partial t} = -\Delta(t)\varphi(u, t), \tag{S2}$$

where $\Delta(t)$ is a time-variant Laplacian operator. Let the state transition matrix be $\Phi_\Delta(t; 0)$ so that the solution of (S2) is given by $\varphi(u, t) = \Phi_\Delta(t; 0)\varphi(u, 0)$ [13].

In [14], the authors discuss the stability radius of time-variant systems. The system still preserves stability if the perturbation of $\Delta(t)$ is smaller than a stability radius. In this discussion, we assume the perturbation is small, i.e., it is inside the stability radius so that both $\Delta(t)$ and its perturbed version $\tilde{\Delta}(t)$ generate exponentially stable solutions. In other words, for a given matrix norm $\|\cdot\|$, there exist positive constants $M$, $\tilde{M}$, $\omega$ and $\tilde{\omega}$ such that

$$\|\Phi_\Delta(t, s)\| \leq M e^{-\omega(t-s)}, \text{ and } \|\Phi_{\tilde{\Delta}}(t, s)\| \leq \tilde{M} e^{-\tilde{\omega}(t-s)}, \quad t \geq s \geq 0. \tag{S3}$$

To simplify the analysis, we further assume the perturbation is small enough such that the exponent difference is also small, i.e.,

$$|\omega - \tilde{\omega}| < \omega. \tag{S4}$$

We now consider the solution difference after perturbation. Let $\mathbf{D}(t)$ and $\mathbf{W}(t)$ be the time-variant version of $\mathbf{D}$ and $\mathbf{W}$, respectively. They are assumed to satisfy (S3) and (S4).

**Lemma S1.** *Let* $\Delta(t) = \mathbf{D}^{-1/2}(t)(\mathbf{D}(t) - \mathbf{W}(t))\mathbf{D}^{-1/2}(t)$ *and* $\tilde{\Delta}(t) = \tilde{\mathbf{D}}^{-1/2}(t)(\tilde{\mathbf{D}}(t) - \tilde{\mathbf{W}}(t))\tilde{\mathbf{D}}^{-1/2}(t)$ *with* $\tilde{\mathbf{D}}(t)$ *being the diagonal degree matrix for* $\tilde{\mathbf{W}}(t) = \mathbf{W}(t) + \mathbf{E}(t)$*, where* $\{\tilde{\mathbf{W}}(t) : t \geq 0\}$ *satisfies* (S3) *and* (S4)*. Denote* $\varepsilon(t) = \|\mathbf{E}(t)\|$*. We have* $\|\varphi(u, t) - \tilde{\varphi}(u, t)\| = O\left(e^{-\rho t} \int_0^t \varepsilon(\tau) \, d\tau\right)$ *for some constant* $\rho > 0$*.*

*Proof.* For the perturbed system, we have

$$\frac{\partial \varphi(u, t)}{\partial t} = -\tilde{\Delta}(t)\varphi(u, t)$$

$$= -\Delta(t)\varphi(u, t) + \left(\Delta(t) - \tilde{\Delta}(t)\right)\varphi(u, t). \tag{S5}$$

Formally, (S5) can be interpreted as a closed loop system obtained by applying the dynamical feedback $\left(\Delta(t) - \tilde{\Delta}(t)\right)\varphi(u, t)$. According to [13], the solution of the above system, denoted by

$\tilde{\varphi}(u, t)$, satisfies

$$\tilde{\varphi}(u, t) = \Phi_\Delta(t, 0)\varphi(u, 0) + \int_0^t \Phi_\Delta(t, \tau)\left(\Delta(\tau) - \tilde{\Delta}(\tau)\right)\tilde{\varphi}(u, \tau)\,\mathrm{d}\tau$$

$$= \varphi(u, t) + \int_0^t \Phi_\Delta(t, \tau)\left(\Delta(\tau) - \tilde{\Delta}(\tau)\right)\tilde{\varphi}(u, \tau)\,\mathrm{d}\tau.$$

Similar to the proof in Section S8.1 for the time-invariant case, we have $\|\!|\Delta(\tau) - \tilde{\Delta}(\tau)|\!\| = O(\varepsilon(t))$. It follows that

$$\|\tilde{\varphi}(u, t) - \varphi(u, t)\| = \left\|\int_0^t \Phi_\Delta(t, \tau)\left(\Delta(\tau) - \tilde{\Delta}(\tau)\right)\tilde{\varphi}(u, \tau)\,\mathrm{d}\tau\right\|$$

$$= \left\|\int_0^t \Phi_\Delta(t, \tau)\left(\Delta(\tau) - \tilde{\Delta}(\tau)\right)\Phi_{\tilde{\Delta}}(\tau, 0)\varphi(u, 0)\,\mathrm{d}\tau\right\|$$

$$\leq \int_0^t \|\!|\Phi_\Delta(t, \tau)|\!\|\,\|\!|\Delta(\tau) - \tilde{\Delta}(\tau)|\!\|\,\|\!|\Phi_{\tilde{\Delta}}(\tau, 0)|\!\|\,\mathrm{d}\tau \cdot \|\varphi(u, 0)\|$$

$$\leq \int_0^t M e^{-\omega(t-\tau)}\tilde{M}e^{-\tilde{\omega}(\tau-0)}\varepsilon(\tau)\,\mathrm{d}\tau \cdot \|\varphi(u, 0)\|$$

$$= O\left(e^{-\rho t}\int_0^t \varepsilon(\tau)\,\mathrm{d}\tau\right)$$

for some constant $\rho > 0$. In the second equality, we have used the fact that the initial point for both the unperturbed and perturbed system is $\varphi(u, 0)$, and the last equality follows from (S3) and (S4). □

**Remark S8.1.** *For the sake of simplicity, in this work, we only present theoretical analysis for the time-invariant or time-variant graph Laplacian in Proposition 1 and Lemma S1 for heat diffusion (11). The more complicated case where the graph Laplacian $\Delta$ depends on node features $u$ as implemented in (16) is left for future work. From experiment results, for example, the results shown in Table 3 in the main paper and Table S10 in this supplementary material, the time-invariant case already preserve some robustness as compared to non-PDE GNNs in Table 1 and Table S9, which validates our theoretical analysis. The more robust mean curvature flow and Beltrami flow proposed in the paper are highly non-linear, making a theoretical analysis difficult. However, our analysis for the time-variant case provides some insights as to why robustness is present in these cases, as demonstrated in our experiments.*

## Broader Impact

Our work develops robust GNNs to mitigate the threat of adversarial attacks, which can lead to reliable deployment of automation in various applications like sensor networks, transportation networks, and manufacturing. This may potentially lead to the replacement of repetitive tasks or jobs that are traditionally performed by humans. However, automation and artificial intelligence (AI) can lead to better productivity, efficiency, and cost-effectiveness with an overall increase in societal living standards. By incorporating the ability to defend against adversarial attacks, our research can lead to more secure and robust adoption of AI technologies. However, potential failures and engineering issues remain challenging open problems.