# OpenReview forum: "On the Robustness of Graph Neural Diffusion to Topology Perturbations"
_NeurIPS.cc/2022/Conference — NeurIPS 2022 Accept_

### Official Review · Reviewer_BHjT · 2022-07-08

**Rating:** 6
**Confidence:** 4
**Soundness:** 3 good
**Presentation:** 3 good
**Contribution:** 3 good

**Summary:**

- The authors provide theoretical and empirical insights into the robustness properties of graph neural PDE networks.
- They show that the heat kernel is stable under small perturbations of the Laplace operator and link this to the graph models by viewing a graph as a discretization of a Riemannian manifold
- They introduce a general graph PDE framework and instantiate three different diffusion processes to analyze their robustness (which are already introduced in related work)

**Questions:**

- What of the robustness is attributed to the Lipschitzness of the network? As a Lipschitz constraint can improve robustness itself, an ablation study would be important.
- The Heat kernel e.g. does not perform better than the APPNP or GAT (for SPEIT attack) while the main robustness result is attributed to the stability of the heat kernel. Can you please explain this result.
- Why do you use so many layers? In general, on graphs, not many layers are used so the motivation for the Lipschitz constraint is unclear to me.
- Why do you need more layers for BLEND / GRAND than in their paper?
- For the inductive setting: are the edges to the unseen data for classification removed before training? Otherwise, due to the increased number of layers, there might be information flow in the training phase


**Limitations:**

-

**Strengths And Weaknesses:**

Strengths:
- The theoretical findings on the robustness of the heat kernel are interesting and novel
- It is the first analysis of neural graph PDE model robustness
- As neural graph diffusion gains importance recently a robustness analysis is important to guide the development of these models

Weaknesses:
- The major weakness is that the analysis itself could be more in-depth, i.e. (i) I recommend to analyze failure modes of that robustness? (ii) Are there specific attacks that these models don’t defend well. (iii) I would suggest evaluation more threat models. Even though the chosen threat model represents attacks in the practice of social networks, for analysis it would be still beneficial to see exposure to other threat models and attack types. In particular, it seems unfair that no attack is introduced specifically targeting the new model.
- It is a bit unclear what is proposed by the authors and what is a new contribution on a first glance. It could be written clearer in that regard.
- Some unclear/open aspects regarding the experiments (see also questions below)

---

> ### Author Response · Authors · 2022-08-02
> **Response 1-2 (to Clarify Reviewer 3 BHjT's Weakness  1 and 2)**
>
> __Response 1.__ Thank you for raising these important questions. Our experiments in the submitted version focus mainly on robustness under realistic injection attacks based on graph topology perturbations. The attackers in injection attacks can only inject new nodes but not modify the target nodes directly. We also refer the reviewer to Response 1-(2) to Reviewer 1 for more explanations of our attack settings used in the main paper.
>
> Under injection attacks with different number of injection nodes and edges, we have performed more experiments with various attack strengths. Table R5 shows that the robustness is not significantly affected by the number of nodes and edges injected by attackers as the original is not perturbed in this setting.
>
> To see the robustness break down, we have implemented modification attacks where attackers can directly flip the original graph’s edges and perturb the features of the nodes. In Table R6, we apply the PGD method to perturb node features and randomly flip edges. We observe that
>
> - the robustness performance starts to break down when the 60% of nodes have their features perturbed by ϵ = 0.1 (the value of features is in [-1,1]) and 60% of edges are flipped, and
>
> - as long as feature perturbation is small, the robustness can still be retained even if 80% of nodes have their features perturbed and 80% of edges are flipped.
>
> In summary, from those extensive experiment results, we observe that our models are more robust against topology perturbation than feature perturbation. We also refer the reviewer to Response 1-(2) to Reviewer 1 for an explanation of the difference between injection and modification attacks.
>
> We have also added more experiments under (injection) white-box attack where the attacker knows everything about the target model and thus can attack it directly. As shown in Table R4, in Response 2 to Reviewer 2, our method maintains good robustness even under white-box attacks.
>
>
> __Response 2.__ Thank you for your suggestion. We have now modified the contribution discussion to clearly state that our main focus is on the robustness to topology perturbation.

---

> > ### Author Response · Authors · 2022-08-02
> > **Table R5(a)**
> >
> > | Dataset | Attack | \# nods/edges injected | Beltrami         | RobustGCN         | GNNGuard         | GCNSVD            | GAT               | GraphSAGE         | GIN              | APPNP             |
> > |---------|--------|------------------------|------------------|-------------------|------------------|-------------------|-------------------|-------------------|------------------|-------------------|
> > |   Cora  | SPEIT  | 50/50                  | 61.87 $\pm$ 0.49 | 36.16 $\pm$ 0.41  | 78.50 $\pm$ 2.27 | 37.50 $\pm$ 0.74  | 38.10 $\pm$ 2.48  | 35.82 $\pm$ 0.01  | 35.82 $\pm$ 0.01 | 36.79 $\pm$ 0.61  |
> > |         | SPEIT  | 100/100                | 57.18 $\pm$ 1.68 | 35.82 $\pm$ 0.00  | 79.39 $\pm$ 1.41 | 35.82 $\pm$ 0.00  | 37.22 $\pm$ 1.87  | 35.82 $\pm$ 0.00  | 35.82 $\pm$ 0.00 | 35.82 $\pm$ 0.00  |
> > |         | SPEIT  | 150/150                | 55.88 $\pm$ 0.56 | 35.82 $\pm$ 0.00  | 80.78 $\pm$ 0.71 | 35.82 $\pm$ 0.00  | 26.12 $\pm$ 16.29 | 35.82 $\pm$ 0.00  | 35.82 $\pm$ 0.00 | 29.85 $\pm$ 11.94 |
> > |         | SPEIT  | 200/200                | 58.21 $\pm$ 2.81 | 35.82 $\pm$ 0.00  | 79.38 $\pm$ 0.64 | 29.88 $\pm$ 11.96 | 29.85 $\pm$ 11.94 | 35.82 $\pm$ 0.00  | 35.82 $\pm$ 0.00 | 22.30 $\pm$ 15.83 |
> > |         | TDGIA  | 50/50                  | 62.84 $\pm$ 1.17 | 53.28 $\pm$ 8.61  | 78.92 $\pm$ 1.80 | 40.77 $\pm$ 3.34  | 35.64 $\pm$ 12.91 | 39.78 $\pm$ 6.46  | 39.63 $\pm$ 2.38 | 60.52 $\pm$ 4.43  |
> > |         | TDGIA  | 100/100                | 62.44 $\pm$ 1.56 | 54.11 $\pm$ 3.18  | 77.99 $\pm$ 2.24 | 42.29 $\pm$ 0.43  | 36.94 $\pm$ 16.96 | 34.24 $\pm$ 9.99  | 37.03 $\pm$ 1.59 | 62.50 $\pm$ 3.47  |
> > |         | TDGIA  | 150/150                | 64.18 $\pm$ 0.87 | 52.43 $\pm$ 10.84 | 79.39 $\pm$ 1.74 | 11.94 $\pm$ 0.00  | 39.74 $\pm$ 16.09 | 38.34 $\pm$ 1.48  | 35.82 $\pm$ 0.00 | 61.37 $\pm$ 7.07  |
> > |         | TDGIA  | 200/200                | 61.48 $\pm$ 1.68 | 48.88 $\pm$ 6.60  | 80.78 $\pm$ 1.89 | 11.94 $\pm$ 0.00  | 46.74 $\pm$ 12.70 | 38.34 $\pm$ 2.50  | 36.19 $\pm$ 0.75 | 61.10 $\pm$ 0.98  |
> > | Citeseer| SPEIT  | 50/50                  | 65.52 $\pm$ 2.26 | 28.56 $\pm$ 7.87  | 69.72 $\pm$ 1.84 | 21.16 $\pm$ 1.32  | 26.00 $\pm$ 11.14 | 19.75 $\pm$ 1.82  | 23.54 $\pm$ 5.30 | 22.19 $\pm$ 0.86  |
> > |         | SPEIT  | 100/100                | 64.74 $\pm$ 1.27 | 17.01 $\pm$ 6.05  | 69.51 $\pm$ 1.61 | 19.75 $\pm$ 2.56  | 19.12 $\pm$ 1.47  | 18.81 $\pm$ 2.74  | 20.61 $\pm$ 2.39 | 20.53  $\pm$ 1.57 |
> > |         | SPEIT  | 150/150                | 65.13 $\pm$ 2.27 | 20.69 $\pm$ 1.49  | 70.22 $\pm$ 1.63 | 18.18 $\pm$ 1.81  | 18.97 $\pm$ 1.57  | 18.34 $\pm$ 1.63  | 19.75 $\pm$ 0.00 | 19.75 $\pm$ 0.00  |
> > |         | SPEIT  | 200/200                | 64.11 $\pm$ 0.83 | 15.52 $\pm$ 7.19  | 69.36 $\pm$ 1.29 | 18.96 $\pm$ 3.00  | 19.12 $\pm$ 1.26  | 22.10 $\pm$ 5.11  | 21.16 $\pm$ 4.97 | 17.71 $\pm$ 1.37  |
> > |         | TDGIA  | 50/50                  | 65.77 $\pm$ 1.28 | 38.81 $\pm$ 10.84 | 69.50 $\pm$ 1.86 | 20.77 $\pm$ 2.52  | 19.63 $\pm$ 6.53  | 28.77 $\pm$ 7.73  | 28.65 $\pm$ 5.08 | 54.48 $\pm$ 8.56  |
> > |         | TDGIA  | 100/100                | 65.60 $\pm$ 1.68 | 45.61 $\pm$ 6.75  | 66.22 $\pm$ 5.44 | 25.39 $\pm$ 6.33  | 25.08 $\pm$ 10.21 | 35.81 $\pm$ 11.99 | 29.94 $\pm$ 7.26 | 52.74 $\pm$ 6.48  |
> > |         | TDGIA  | 150/150                | 66.22 $\pm$ 2.31 | 39.66 $\pm$ 15.80 | 69.36 $\pm$ 0.90 | 24.14 $\pm$ 3.21  | 19.36 $\pm$ 2.14  | 26.57 $\pm$ 9.28  | 27.51 $\pm$ 6.69 | 59.01 $\pm$ 3.93  |
> > |         | TDGIA  | 200/200                | 64.66 $\pm$ 2.15 | 38.48 $\pm$ 9.63  | 69.91 $\pm$ 1.54 | 26.49 $\pm$ 4.14  | 22.33 $\pm$ 11.29 | 26.72 $\pm$ 13.41 | 26.73 $\pm$ 7.47 | 51.72 $\pm$ 5.06  |
> >
> > __Table R5__: Node classification accuracy (\%) on adversarial examples generated from SPETI and TDGIA under black-box injection setting where different number of injected nodes and edges are applied.

---

> > > ### Author Response · Authors · 2022-08-02
> > > **Table R5(b)**
> > >
> > > | Dataset | Attack | \# nodes/edges injected | Beltrami         | RobustGCN         | GNNGuard         | GCNSVD            | GAT               | GraphSAGE         | GIN              | APPNP             |
> > > |---------|--------|------------------------|------------------|-------------------|------------------|-------------------|-------------------|-------------------|------------------|-------------------|
> > > |  PubMed | SPEIT  | 300/300                | 86.66 $\pm$ 0.68 | 75.54 $\pm$ 0.54  | 84.36 $\pm$ 0.58 | -                 | 40.94 $\pm$ 2.47  | 39.22 $\pm$ 0.00  | 76.71 $\pm$ 0.14 | 77.55 $\pm$ 0.54  |
> > > |         | SPEIT  | 600/600                | 86.80 $\pm$ 0.98 | 75.71 $\pm$ 0.60  | 86.47 $\pm$ 0.58 | -                 | 34.8 $\pm$ 10.99  | 41.10 $\pm$ 1.63  | 76.36 $\pm$ 0.65 | 77.22 $\pm$ 0.43  |
> > > |         | SPEIT  | 900/900                | 87.10 $\pm$ 0.38 | 76.00 $\pm$ 0.35  | 86.73 $\pm$ 0.48 | -                 | 39.99 $\pm$ 1.55  | 40.78 $\pm$ 1.77  | 76.73 $\pm$ 0.36 | 77.14 $\pm$ 0.70  |
> > > |         | SPEIT  | 1200/1200              | 87.51 $\pm$ 0.58 | 74.75 $\pm$ 0.90  | 86.19 $\pm$ 0.40 | -                 | 40.82 $\pm$ 1.85  | 40.08 $\pm$ 1.63  | 76.70 $\pm$ 0.72 | 77.17 $\pm$ 0.32  |
> > > |         | TDGIA  | 300/300                | 85.56 $\pm$ 0.91 | 75.53 $\pm$ 0.36  | 84.00 $\pm$ 1.12 | -                 | 39.78 $\pm$ 0.29  | 60.40 $\pm$ 11.23 | 77.58 $\pm$ 0.71 | 77.45 $\pm$ 0.68  |
> > > |         | TDGIA  | 600/600                | 86.33 $\pm$ 0.84 | 75.84 $\pm$ 0.23  | 84.21 $\pm$ 0.33 | -                 | 56.00 $\pm$ 9.13  | 70.24 $\pm$ 20.07 | 76.38 $\pm$ 0.75 | 77.37 $\pm$ 0.31  |
> > > |         | TDGIA  | 900/900                | 87.00 $\pm$ 1.47 | 75.91 $\pm$ 0.55  | 83.87 $\pm$ 1.01 | -                 | 41.44 $\pm$ 10.61 | 52.40 $\pm$ 6.03  | 75.56 $\pm$ 0.58 | 76.88 $\pm$ 0.68  |
> > > |         | TDGIA  | 1200/1200              | 86.87 $\pm$ 1.41 | 75.62 $\pm$ 0.82  | 84.35 $\pm$ 2.24 | -                 | 57.16 $\pm$ 10.92 | 71.72 $\pm$ 14.02 | 76.49 $\pm$ 0.80 | 77.23 $\pm$ 0.46  |
> > >
> > >
> > >
> > >
> > > __Table R5__: Node classification accuracy (\%) on adversarial examples generated from SPETI and TDGIA under black-box injection setting where different number of injected nodes and edges are applied.

---

> ### Author Response · Authors · 2022-08-02
> **Response 3 (to Clarify Reviewer 3 BHjT's Questions Section)**
>
> __Response 3.__
>
> (1). Thank you for your suggestion. We now include an ablation study with and without the Lipschitz constraint. Table R7 shows the performance of our method when there is no Lipschitz constraint. In summary, the Lipschitz constraint does slightly improve the clean accuracy and it also improves robustness by 2 ∼ 5%. We have also included this study in the revision.
> | Clean/Robust acc. | \# layers used |     Beltrami     | Beltrami w/o Lips. |     |     |
> |:-----------------:|:--------------:|:----------------:|:------------------:|:---:|:---:|
> |       Clean       |       1        | 63.96 $\pm$ 1.95 |  63.51 $\pm$ 3.18  |     |     |
> |                   |       2        | 72.46 $\pm$ 0.76 |  69.93 $\pm$ 1.48  |     |     |
> |                   |       3        | 73.51 $\pm$ 1.49 |  72.69 $\pm$ 1.89  |     |     |
> |                   |       4        | 75.93 $\pm$ 1.46 |  75.15 $\pm$ 1.41  |     |     |
> |      Robust       |       1        | 59.10 $\pm$ 3.20 |  58.06 $\pm$ 1.78  |     |     |
> |                   |       2        | 63.58 $\pm$ 1.39 |  58.96 $\pm$ 2.82  |     |     |
> |                   |       3        | 60.52 $\pm$ 1.56 |  57.46 $\pm$ 1.65  |     |     |
> |                   |       4        | 61.87 $\pm$ 0.49 |  56.79 $\pm$ 1.52  |     |     |
> |                   |                |                  |                    |     |     |
>
> __Table R7__: Node classification accuracy (%) on adversarial examples generated from
> SPEIT. We apply graph neural PDEs induced from Beltrami flow with or
> without the Lipschitz constraint. Experiments are conducted on Cora
> dataset.
>
> (2). Thank you for your insightful question. "Heat" in Table 3 serves as a baseline. As we explained in line 271-272, both "Heat" and GRAND/BLEND are derived from heat flow, where GRAND/BLEND uses the attention function to weigh edges while "Heat" uses the constant function, which is not learned from the data and hence not expected to perform well. In a further experiment, we now treat the constant in "Heat" as a trainable variable which is shared by all edges. We denote this new variant as Heat + and compare it with GAT and APPNP under SPEIT attack in Table R8. We see that Heat+  is more robust. This constant variable controls the diffusion diffusivity in graph, analogous to the thermal diffusivity in manifold. It is interesting to observe such a phenomenon since the experiment indicates that heat diffusivity also affects the robustness. We believe further investigations in this direction can be performed in future work.
>
> (3). We applied three layers in most of the cases (with a few exceptions that used four layers). We believe using three or four layers is common for most of the existing GNNs without causing over-smoothing and/or over-squashing problems.
>
> We agree that fewer layers can mitigate over-smoothing problem. However, we found that when using just one or two layers, the model is somewhat under-fitted, i.e., the test clean accuracy is low. To understand this better, we performed experiments using different number of layers. The results are summarised in Table R7. The number of layers to use is a hyperparameter we tune during training. We have also clarified this in our revision.
>
>
> (4). We have tried using the same number of layers for GRAND/BLEND as in their paper. However, the test clean accuracy is low. This is mainly due to two reasons: 1) we are using inductive training whereas the paper of GRAND/BLEND uses transductive training; 2) the datasets we use have been calibrated for this robustness evaluation, which is different from the original datasets used by the paper of GRAND/BLEND. For example, grb-cora [R3.1] has node feature size of 302 while the original Cora dataset has node feature size of 1433. We have clarified this in our revision.
>
> [R3.1] Q. Zheng, X. Zou, Y. Dong, Y. Cen, D. Yin, J. Xu, Y. Yang, and J. Tang, “Graph robustness benchmark: Benchmarking the adversarial robustness of graph machine learning,” in Proc. Advances Neural Inf. Process. Syst. Track on Datasets and Benchmarks, 2021.
>
> Table R7 provides empirical evidence for GRAND/BLEND needing more layers in our setting.
>
> (5). Yes, unseen data are masked out during training. We have pointed this out explicitly in our revision.

---

> > ### Author Response · Authors · 2022-08-02
> > **Table R8**
> >
> >
> > | Dataset  |   GNN    |      Clean       |      10/10       |      15/15       |       20/20       |      25/25       |      30/30       |       35/35       |       40/40       |      45/45       |       50/50       |
> > |:--------:|:--------:|:----------------:|:----------------:|:----------------:|:-----------------:|:----------------:|:----------------:|:-----------------:|:-----------------:|:----------------:|:-----------------:|
> > | Citeseer | Heat$^+$ | 69.78 $\pm$ 0.95 | 69.09 $\pm$ 1.61 | 67.77 $\pm$ 1.61 | 66.27 $\pm$ 2.41  | 66.27 $\pm$ 2.81 | 64.26 $\pm$ 1.27 | 61.88 $\pm$ 3.01  | 57.30 $\pm$ 4.07  | 54.61 $\pm$ 5.25 | 54.61 $\pm$ 2.92  |
> > |          |   GAT    | 69.81 $\pm$ 1.43 | 60.56 $\pm$ 4.77 | 62.01 $\pm$ 4.89 | 45.52 $\pm$ 10.44 | 30.34 $\pm$ 5.40 | 31.16 $\pm$ 6.16 | 31.03 $\pm$ 12.60 | 36.11 $\pm$ 13.85 | 23.13 $\pm$ 3.25 | 26.00 $\pm$ 11.14 |
> > |          |  APPNP   | 70.75 $\pm$ 0.86 | 68.65 $\pm$ 0.91 | 66.52 $\pm$ 2.90 | 59.87 $\pm$ 3.49  | 56.11 $\pm$ 9.70 | 56.05 $\pm$ 2.37 | 51.22 $\pm$ 8.88  | 38.50 $\pm$ 8.75  | 31.60 $\pm$ 9.80 | 22.19 $\pm$ 0.86  |
> >
> > __Table R8__: Node classification accuracy (%) on adversarial examples generated from
> > SPEIT, when the number of injected nodes/edges is varying.

---

> ### Author Response · Authors · 2022-08-08
> **To Reviewer BHjT**
>
> Dear Reviewer BHjT, we would like to express our thanks for your helpful comments which have definitely made our paper better! Could we kindly ask whether our point-by-point responses and the uploaded paper revision have addressed your main concerns about our work? Thank you again for your efforts in raising valuable questions and improving our paper!

---

> > ### Comment · Reviewer_BHjT · 2022-08-09
> > **Reply**
> >
> > I thank the authors for their feedback and changes / additional experiments addressing my comments. It is good that the authors frame their work to be mainly focusing on the topology.
> >
> > Overall I think the paper discusses an interesting and important topic but I still feel that the evaluation could be more thorough. This would include an extensive discussion. However, the approach attributing the intrinsic robustness to the heat kernel is good. A few further comments:
> >
> > - Compare the PGD attacks with the other GNNs / defense methods (also better in a figure instead of a table)
> > - Table R5 (a) and (b) show that SPEIT and TDGIA are not really depending on the nodes/edges injected. The authors should include more options to see the tipping point for the most models (e.g. GCNSVD already broke down on 50/50
> > - The new experiments in R5 raise questions about the usefulness of TDGIA for these experiments: the GCN as surrogate model also seems to not be able to attack APPNP showing that it might be just the difference between surrogate model (GCN) and target model increasing robustness in this setting
> > - Include other attacks (also injection) and focus on differences in the white-box attacks if they are fair (with respect to gradient masking etc.)
> >
> > I will raise the score to 6.

---

> > > ### Author Response · Authors · 2022-08-09
> > > **Response to Reply**
> > >
> > > Dear Reviewer BHjT, thank you for the new comments. The new proposed evaluations are valuable and it is indeed reasonable to conduct them. Due to the tight timelines, we do not have enough time to run the new experiments during this rebuttal period which ends in several hours. However, we have marked down them and will do more evaluations according to the comments. We will include the results in the new paper version.
> > >
> > > Thank you again for your enormous efforts in the reviewing process. Your great comments indeed help us improve the experimental evaluations of our work a lot!

---

### Official Review · Reviewer_w639 · 2022-07-11

**Rating:** 7
**Confidence:** 2
**Soundness:** 3 good
**Presentation:** 3 good
**Contribution:** 3 good

**Summary:**

The authors address the important property of robustness to adversarial attacks in the context of graph neural PDEs. This is the first work to do so. The authors demonstrate that graph neural PDEs are more robust than other GNNs. The authors propose a theoretical reason for this robustness property.

**Questions:**

How do lines 92-94 follow from [31, theorem 7.1]? Perhaps it should be theorem 7.13?

Would Figure 1 make sense/be easier to compare if you used t-SNE on one instance and then used the same position to plot the other graphs? (similarly, figure 3).

The codebase is huge. Has it been forked from an existing repo?

**Limitations:**

I address the main limitations in the strengths/weaknesses section.

**Strengths And Weaknesses:**

The paper is well written and well organised.

The authors show that heat diffusion on a Riemannian manifold is robust to small perturbations of the manifold metric. The authors use this as a justification for why graph neural PDEs are robust to changes in the underlying graph. I think the authors should justify why small perturbations to the underlying graph give small perturbations to the graph metric. This seems to be assumed in lines 186-189. In the case of the shortest path distance, this can change quite radically with just a single edge removal/addition. As an example, consider a cycle graph. By removing an edge, the distance between nodes close to the removed edge goes from very small to very large. By increasing the size of the graph, you can make this change arbitrarily drastic.

The experiments are extensive and convincing. However, the black box attacks trains a surrogate model which is closer to many of the benchmark models (message passing GNNs). This may give an unfair advantage to the method presented in the paper.

“Table 2 shows that BeltramiGuard makes attacks’ efforts totally in vain and exceeds GNNGuard in Citeseer and PubMed datasets.”  The citeseer results are very similar (specially given the error on table 1 for GNNGuard). Can the authors check table 2 is correct for Citeseer, as the values are identical.

The datasets used are all very small, it may be more relevant to try a larger dataset although given the large combination of models/dataset this could be forgiven on computational grounds. Is it feasible to run experiments on a larger dataset in the review period?

Minor comments:
- Table 1 should be centred.
- The paper would be more self-contained and easier to read if the authors defined symmetric measurable coefficients used in Theorem 1.
- The codebase is quite complex with just a small README.md It would be useful to have instructions to reproduce the main figures/tables in the paper.

---

> ### Author Response · Authors · 2022-08-02
> **Response 1-5 (to Clarify Reviewer 2 w639's Weaknesses and Minor Comments)**
>
> __Response 1.__  The addition or removal of an edge can change geodesic distances dramatically over a large portion of the graph. However, the stability of the heat kernel comes from its interpretation as the transition probability of Brownian motion on the manifold. Intuitively, this means that $k_t (x,y)$ is a weighted average over __all__ possible paths between points (nodes) $x$ and $y$ at time $t$, which should not be greatly affected by local perturbations of the manifold (graph). Large changes in  geodesic distances do **not** necessarily imply large heat kernel perturbations. We refer the reviewer to [R2.1, Figure 7] for an experiment where geodesic distance changes between a large portion around two points of the manifold when a small tunnel is added to the manifold to connect these two points, while the heat diffusion still remains similar compared to diffusion under the structure of the original manifold.
>
> Regarding the cycle graph example proposed by the reviewer, the removal of the edge will change the two endpoints of the removed edge dramatically. However, the other points remain stable especially for the points away from the two endpoints since the kernel $k_t (x,y)$ is not dramatically affected there as explained above.
>
> The robustness becomes even more clear if we consider a lattice graph. One additional edge to connect two far away nodes will not dramatically change the heat kernel, but the geodesic distances change dramatically, similar to the experiment conducted in [R2.1]. Furthermore, according to our Proposition 1., if the cycle graph is a weighted graph and the removed edge in the cycle graph is small, the diffusion will not be affected much since the the perturbation $||\mathbf{E}||$ is small.
>
>
> [R2.1] J. Sun, M. Ovsjanikov, and L. Guibas, “A concise and provably informative multi-scale signature
> based on heat diffusion,” Computer graphics forum, vol. 28, no. 5, pp. 1383–1392, 2009.
>
>
> __Response 2.__  We now include the white box attacks for our model and the baselines. The results are shown in Table R4. We observe that our model outperforms the baselines under white box attacks. (We have also included the white box experiments in the revision.)
>
> __Response 3.__ The values in Table 2 in the paper are correct. The scores for Citeseer are identical since BeltramiGuard has defended the attack 100% successfully, i.e., the accuracy under the attack models are the same as the clean accuracy.
>
> __Response 4.__ We included three medium size dataset in the supplementary material. We refer the reviewer to Table S1 in the supplementary material for the dataset details. For large-scale dataset such as ogbn-arxiv, our GPU memory is insufficient for running graph neural PDEs. This is the limitation of neural PDE solvers. To improve scalability, one possible way is to use sample-then-aggregate approaches (similar to what GraphSAGA does). I.e., we may apply local neural PDEs on the sampled subgraphs and then aggregate the PDE outputs. We leave this exploration for a future work.
>
> __Response 5.__ Regarding the Minor comments:
>
> (1). Thank you. We have centered Table 1 in the revision.
>
> (2). We have explicitly defined the symmetric measurable coefficients before Theorem 1 in the revision. In the previous submission version, we mention them around line 98.
>
> (3). The code will be released to Github with more detailed instructions. We apologize for the current unclear instructions in the codebase.

---

> > ### Author Response · Authors · 2022-08-02
> > **Table R4**
> >
> >
> > | Dataset  | Attack  |     BeltramiGuard      |        Beltrami      |       RobustGCN        |        GNNGuard        |      GCNSVD      |       GAT        |       GraphSAGE        |        GIN        |        APPNP         |
> > |:--------:|:-------:|:----------------------:|:----------------------:|:----------------------:|:----------------------:|:----------------:|:----------------:|:----------------------:|:-----------------:|:--------------------:|
> > |   Cora   | *clean* |    73.01 $\pm$ 2.01    |    75.93 $\pm$ 1.46    | ***81.34 $\pm$ 0.66*** |    79.44 $\pm$ 1.18    | 69.28 $\pm$ 1.37 | 79.74 $\pm$ 1.59 |    76.75 $\pm$ 1.52    | 76.79 $\pm$ 1.35  | **83.06 $\pm$ 1.06** |
> > |          |  SPEIT  | ***71.94 $\pm$ 0.31*** |    48.95 $\pm$ 3.32    |    36.12 $\pm$ 0.31    |  **80.22 $\pm$ 0.91**  | 33.06 $\pm$ 7.86 | 19.18 $\pm$ 9.47 |   17.16 $\pm$ 11.30    | 21.49 $\pm$ 13.08 |  19.77 $\pm$ 10.39   |
> > |          |  TDGIA  | ***67.39 $\pm$ 1.97*** |    52.61 $\pm$ 4.20    |    36.27 $\pm$ 0.61    |  **78.43 $\pm$ 1.10**  | 11.94 $\pm$ 0.0  | 7.99 $\pm$ 4.40  |    21.57 $\pm$ 4.82    | 38.06 $\pm$ 4.69  |   51.94 $\pm$ 5.32   |
> > | Citeseer | *clean* |    69.90 $\pm$ 0.44    | ***70.41 $\pm$ 1.38*** |    70.72 $\pm$ 1.15    |    69.69 $\pm$ 1.83    | 66.93 $\pm$ 1.07 | 69.81 $\pm$ 1.43 |    69.78 $\pm$ 1.31    | 68.81 $\pm$ 1.58  | **70.75 $\pm$ 0.86** |
> > |          |  SPEIT  | ***68.78 $\pm$ 0.82*** |    55.24 $\pm$ 6.90    |    20.19 $\pm$ 2.61    |  **69.22 $\pm$ 1.90**  | 19.31 $\pm$ 3.55 | 14.67 $\pm$ 5.05 |    19.81 $\pm$ 3.06    | 12.54 $\pm$ 6.33  |   20.75 $\pm$ 2.32   |
> > |          |  TDGIA  | ***67.96 $\pm$ 1.22*** |53.61 $\pm$ 14.01|    18.68 $\pm$ 4.06    |  **68.40 $\pm$ 1.44**  | 16.93 $\pm$ 2.41 | 14.23 $\pm$ 5.05 |    20.69 $\pm$ 5.64    | 18.87 $\pm$ 3.61  |   25.70 $\pm$ 6.78   |
> > |  PubMed  | *clean* |    87.77 $\pm$ 0.14    |  **86.94 $\pm$ 0.25**  |    75.55 $\pm$ 0.32    |    84.80 $\pm$ 0.51    |        \-        | 84.91 $\pm$ 0.76 | ***89.22 $\pm$ 0.25*** | 76.71 $\pm$ 0.14  |   77.50 $\pm$ 0.54   |
> > |          |  SPEIT  |  **85.26 $\pm$ 1.42**  | ***85.13 $\pm$ 0.79*** |    75.07 $\pm$ 0.30    |    84.30 $\pm$ 1.34    |        \-        | 39.22 $\pm$ 0.0  |    40.46 $\pm$ 1.69    | 75.58 $\pm$ 1.03  |   77.62 $\pm$ 0.10   |
> > |          |  TDGIA  |    81.36 $\pm$ 3.09    |  **84.88 $\pm$ 0.46**  |    75.78 $\pm$ 0.32    | ***83.33 $\pm$ 2.91*** |        \-        | 37.96 $\pm$ 1.82 |    44.85 $\pm$ 3.06    | 75.72 $\pm$ 0.70  |   77.32 $\pm$ 0.44   |
> >
> > __Table R4__: Node classification accuracy (%) on adversarial examples generated from
> > *white-box* attacks. The best and the second-best result for each
> > criterion are highlighted in **bold** and ***bold and italic***
> > respectively.

---

> > ### Comment · Reviewer_w639 · 2022-08-08
> > **Comment on response 1**
> >
> > I thank the authors for their response. They provide a thorough response to my comments.
> >
> > Further comments on response 1: Thank you for expanding on this. The response makes sense. However, I can still imagine scenarios where small perturbations (in terms of the number of edges changed) can cause a large perturbation in this context. As an example, consider a graph with three communities (A, B and C) which are sparsely connected (say, an SBM graph). Let x be a node in A and y a node in C. Then, by deleting the few existing edges connecting communities A and C, the shortest paths between x and y that pass between these communities will disappear, leaving the longer paths that pass through B. This change is true for many pairs of nodes (i.e. all combinations of nodes where one is in A and one is in C). I agree with the authors that this "should not be greatly affected by local perturbations of the manifold (graph)" for most practical cases. However, the perturbations considered are adversarial, so it is reasonable that all cases should be covered. For this reason, I think a theoretical justification that small geodesic distances imply small heat kernel perturbations could improve this work.

---

> > > ### Author Response · Authors · 2022-08-08
> > > **Response to the new comment on response 1**
> > >
> > > Thank you very much for the new provided example that may help the picture more clear about the relation between heat kernel perturbation and the geodesic perturbation.
> > >
> > > In the current work in Proposition 1, we directly assume the few edge perturbations will make the metric perturbation small, and therefore the heat kernel perturbation is small. For small changes in geodesic distance, we then know the metric perturbation is small (since geodesic distance perturbation between any two points is small) and again the heat kernel perturbation is small. For large changes in geodesic distance, we can only conclude large changes in geodesic distance __do not necessarily__ imply large heat kernel perturbations and therefore do not necessarily imply large solution perturbations. The example you proposed is highly non-trivial. The removal of edges between $A$ and $C$ will force large geodesic distance perturbations. However, large geodesic distance perturbations  __still do not necessarily imply large solution perturbations__. In the example of three communities as mentioned above, removing the few existing edges connecting communities $A$ and $C$ may not affect the downstream task such as node classification: if all the nodes in the same community belong to the same class, then removing inter-community connection which largely affects the geodesic distance between the nodes from $A$ and $C$, however, has little impact on the node classification results. Of course, we can always find a counter-example where removing an edge (or a few edges) that largely affects both the geodesic distance between the associated nodes and the downstream task's accuracy (in this case adversaries can flip edges that result in large geodesic distance change).
> > >
> > > The theoretical work between geodesic distance perturbation and solution perturbation is therefore highly non-trivial. We think it may depend on other geometric properties like the Balanced Forman curvature of a graph [R2.2] since the heat diffusion process on a manifold tends to diffuse differently under different geometries. For example, it diffuses slower at points with positive curvature, and faster at points with negative curvature. From experiments conducted in Table R8, we also observe that the diffusion diffusivity in a graph (analogous to the thermal diffusivity in a manifold) affects the model robustness. Furthermore, in the literature on parabolic equations, there exist unstable __singular__ (not heat diffusion) parabolic equations [R2.3] for which the solution may blow up in finite time. This unstable phenomenon may also be related to the stability of more general graph neural partial differential equations.
> > > We point out the possible theoretical work directions and will study them in the next coming months (or even years).
> > >
> > > Finally, we would like to sincerely thank you again for your enormous efforts putting in reviewing our paper and raising valuable questions! This has improved our current work a lot as shown in the revision and also inspires possible future research directions!
> > >
> > > [R2.2] J. Topping, F. Di Giovanni, B. P. Chamberlain, X. Dong, and M. M. Bronstein, “Understanding over-squashing and bottlenecks on graphs via curvature,” in Proc. Int. Conf. Learn. Representations, 2022.
> > >
> > > [R2.3] E. DiBenedetto, Degenerate parabolic equations. Springer Science & Business Media, 1993.

---

> > > > ### Comment · Reviewer_w639 · 2022-08-09
> > > > **Response**
> > > >
> > > > I thank the authors for their detailed response. In particular, the second paragraph describes the non-trivial connection between geodesic distance perturbation and solution perturbation. The authors have provided an extensive and detailed discussion and answered all the questions I had about the work and I thank them for the interesting discussion. I would also like to acknowledge the improvements made by including experiments with white-box attacks, I think this makes for a more convincing experimental section. I have updated my score according to the improvements made during the rebuttal period.

---

> > > > > ### Author Response · Authors · 2022-08-09
> > > > > **thanks to Reviewer w639**
> > > > >
> > > > > Thank you sincerely again for your enormous efforts and time spent in the reviewing process. Your comments indeed help us improve our work a lot!

---

> ### Author Response · Authors · 2022-08-02
> **Response 6 (to Clarify Reviewer 2 w639's Questions Section)**
>
>
> __Response 6.__  Regarding the __Questions__ section:
>
> (1). Sorry for this typo. Yes, it should be Theorem 7.13. We have corrected this typo in the revision.
>
> (2). Thank you for your suggestion. We indeed considered the same thing when we were plotting the t-SNE graphs. The reason why we use the current t-SNE plots in the paper is because of the following.
> The position of each data point in the t-SNE plots are their 2D embeddings learned by t-SNE. That means that even if the same random seed is set for two t-SNE processes, the output (the t-SNE 2D embeddings) will not be the same as the inputs (high-dimensional embeddings learned by the GNNs) to t-SNE are different. If we used t-SNE on one instance and then used the same position to plot the other graphs, the relative similarity between the data points on the other graphs will not be correctly reflected.
>
> (3). Our codes are developed based on the following two repositories:
> https://github.com/twitter-research/graph-neural-pde and
> https://github.com/THUDM/grb,
> where our new diffusion schemes and their induced neural PDEs are developed based on the first repo and we follow the second repo to set up the robustness evaluation benchmark.
>
>
> Our code will be released to Github with more details and citations.

---

> > ### Comment · Reviewer_w639 · 2022-08-08
> > **t-SNE**
> >
> > In response to (2) can the authors clarify what features are used to fit the t-SNE embedding? I assumed it was the original node features or labels, in which case it seems reasonable to use the same position for all three plots?

---

> > > ### Author Response · Authors · 2022-08-08
> > > **Response to t-SNE new comments**
> > >
> > > We did not use the original (input) node features but extracted the features from the last layer of a trained model to fit the t-SNE embedding. As compared to the original node features, the model's output features are low-dimensional (easy to fit) and more representative.
> > >
> > > If we only want to visualize the attention weights (on edges), we will take your suggestion to use the input features when plotting t-SNE embedding so that the nodes' positions are consistent over plots. The new plots will be included in our revision.

---

> ### Author Response · Authors · 2022-08-09
> **Any other concerns?**
>
> Dear Reviewer w639, thank you very much again for your helpful comments! Could we kindly ask if you have any other concerns? The Author- Reviewer Discussion will end soon.

---

### Official Review · Reviewer_UKSj · 2022-07-14

**Rating:** 6
**Confidence:** 3
**Soundness:** 3 good
**Presentation:** 3 good
**Contribution:** 3 good

**Summary:**

Through the analysis of heat flow, this paper proposes a more general graph neural diffusion scheme, where the authors induce a new class of PDE-based graph neural networks.
The authors further explore the graph neural PDEs’ robustness properties from a theoretical point of view and show it is Lyapunov stable.
The numerical experiments show certain robustness of the induced graph neural PDEs (including mean curvature flow-based and Beltrami flow-based methods) against topology perturbations.


**Questions:**

The following questions mainly related to the above weaknesses:
1. Are the proposed methods or theoretical analysis suitable for the node attribute to perturb scenario?
2. How do the experiments reflect the setting mentioned in Line 251-258?
3. How long does it need to run the proposed model?

It would be appreciated if the authors could also answer other questions or problems listed in the weakness.


**Limitations:**

Related to the above weakness and Questions, one of the limitations I can see is that the discussion in the paper may not be suitable for all the attack scenarios, such as node attribute perturbation.
Another limitation is that the theory guard may not be valid once the perturbation is significant or bigger than a certain “threshold”. Those two potential limitations may need lots of effort and explorations in the future.

To the best of my knowledge, there is no potential negative societal impact.


**Strengths And Weaknesses:**

Strengths:
1. This paper first reviews a couple of math concepts and notions, including differential geometry, diffusion equations, gradient/divergence operators, etc. Those preliminaries are introduced systematically, and close to the following sections. Therefore, it helps the reader quickly get the main idea of each section instead of being stuck in the learning curve.
2. The paper explores graph neural PDEs' robustness property and mathematically shows they are inherently Lyapunov stable and robust against topology perturbations.
3. The author proposed a model framework and several insights to design a new class of GNNs based on heat flows. The numerical experiments also validate the above analysis to some degree. Lastly, the related codes are also open-sourced, benefiting the community.

Weaknesses:
1. One of my concerns comes from the title and some paragraphs (esp. Line 24-27 and Line 251-258). The paper seems to target the "robustness" topic, which covers a large area (including topology modification, node attribute perturbation, different attack settings, and so on) as the above sentence/paragraphs mentioned in the paper. However, the proposed theoretical analysis (i.e., propositions 1 and 2) and numerical experiments only focus on topology perturbation, which is just a portion of the big robustness topic.
In other words, I am unclear the roles of Line 24-27 and 251-258. For example, Line 251-258 discuss four different attack settings, but I didn't see how the two methods (SPEIT and TDGIA) used in the experiments tie back to those four settings.
2. The experiment results are not well analyzed. They should connect to the theory in the previous section closer. For example, the reason for good or bad performance is not well illustrated as we can see the proposed method performance in Table 1 and Table S2 is not always outperforming others. Moreover, the propositions require a small perturb assumption. It will also be interesting to see more discussions about the impact of different perturb degrees on model performance.
3. Lastly, there are some other minor issues:
(1) It is better if the author can report the training and test speed, or do some algorithm/memory complexity analysis, especially for the section “PDE Solvers”. In this way, the paper can demonstrate better in its practical value.
(2) It would be better if some table can make consistency. For example, Table 4 and Table S2 don’t report the results under the TDGIA setting. And I find the Table 4 results are not match with any other previous tables, so what is the PDE solver used in Table 1?
(3) Since the author mentioned the over-smoothing problem and stacking multiple layers, it would be better if the author could provide more information about it.

---

> ### Author Response · Authors · 2022-08-02
> **Response 1 (to Clarify Reviewer 1 UKSj's Weakness 1)**
>
> __Response 1.__
>
> (1) Sorry for the confusion here. We will revise our paper (abstract, introduction and experimental setup) by clearly stating that we mainly focus on dealing with topology perturbation and our experiments demonstrate the robustness of neural PDEs to topology perturbations. We will change the paper title to “On the Robustness of Graph Neural Diffusion to Topology Perturbations” to make it more explicit.
>
> The following explains how the selected attack methods are related to the attack settings.
>
> (2) There are three common categories of adversarial attacks studied in the literature [R1.6]:
>
> - Poison (attack occurs in training) or evasion (attack occurs in testing).
>
> - White-box (attackers knows target model/method) or black-box (attackers do not know target model/method and thus need to attack a surrogate model and then transfer to target model).
>
> - Injection (attackers inject nodes/edges to the original graph and generate attributes for the injected nodes) or modification (attackers modify the original graph including its topology and node features directly).
>
>
>
> Our experiments in the previous submitted version mainly focus on the evasion, black-box and injection attack setting, which we believe are the most realistic attack settings. The two selected attack methods SPEIT and TDGIA are tailored for these attack settings. To be more specific, we carry out the following:
>
> - Evasion: SPEIT and TDGIA are performed on a trained model during testing time.
>
> - Black-box: SPEIT and TDGIA are used to attack a trained GCN, i.e., a surrogate model, to generate graph perturbations and then the target model is tested on this perturbed graph.
>
> - Injection: when perturbing the graph based a trained GCN, SPEIT and TDGIA firstly inject new nodes into the original graph and then generate the injected nodes’ features.
>
>
>
> The above details are provided in the supplementary material. We also include node attribute perturbation experiments. Please refer to Table R7 and Response 1 to Reviewer 3 for more details.
>
> In this rebuttal, we test additional attack settings. Please refer to our Response 2 to Reviewer 2 and Table R4 which uses evasion, _white-box_ and injection attack. In this setting, SPEIT and TDGIA are used to directly attack the target model instead of a surrogate model to generate graph perturbations.
>
> We have also included the above explanation for experiments in the updated revision.

---

> ### Author Response · Authors · 2022-08-02
> **Response 2 (to Clarify Reviewer 1 UKSj's Weakness 2)**
>
> __Response 2.__
>
> (1). We explain the connection between experiment results and the robustness theory in the revision:
>
> Table 1 in the main paper demonstrates that graph neural PDE induced from Beltrami flow is more robust than all other GNNs under black-box and injection attacks except for GNNGuard, which is specifically designed to remove malicious edges and is thus robust against topology perturbation. This shows that the output features from a graph PDE are stable under topology perturbation, as suggested by Proposition 1.
>
> From experiment results shown in Table 4 in the main paper, we observe that even the vanilla time-invariant heat flow preserves some robustness as compared to non-PDE GNNs in Table 1. This further validates our theoretical analysis in Proposition 1, which suggests that if the topology perturbation is bounded, the learned representations are close to those under the “clean” scenario.
>
> (2). Here, we give some explanation for the model performance in Table 1 and Table S2 in the paper and the supplementary material mentioned by the reviewer. (This has also been included in the revision.)
>
> The heat diffusion process on a manifold tends to diffuse differently under different geometries. For example, it diffuses slower at points with positive curvature, and faster at points with negative curvature. In [R1.4] and [R1.5], the authors show different datasets have different geometric properties like hyperbolicity distribution or Balanced Forman curvature. Our theoretical analysis only shows a loose uniform bound in terms of the adjacency matrix, while the performance is very much dataset dependent. More advanced theoretical analysis for different datasets is highly non-trivial and needs further extensive investigations.
>
> (3). We note that in adversarial attacks, perturbations are assumed to be imperceptible [R1.1], otherwise attack detection techniques can be used [R1.2, R1.3]. Therefore, most literature on robust techniques against adversarial attacks assumes small perturbation. Here, to study the impact of different perturbation degrees on model performance, we include more experiments in Table R5 (injection attack), Table R6 (modification attack) and Table R8 (vanilla heat diffusion under SPEIT attack) where we increase the portion of nodes/edges that are allowed to be perturbed. As expected, when the perturbation is smaller, our model can provide good adversarial robustness.
>
> [R1.1] Szegedy, C., Zaremba, W., Sutskever, I., Bruna, J., Erhan, D., Goodfellow, I., and Fergus, R., "Intriguing properties of neural networks," in Proc. Int. Conf. Learning Representations, 2014.
>
> [R1.2] Meng, Dongyu, and Hao Chen, "Magnet: a two-pronged defense against adversarial examples." in Proc. ACM SIGSAC Conf. Comput. Commun. Secur., 2017.
>
> [R1.3] X. Zhang and M. Zitnik, “Gnnguard: Defending graph neural networks against adversarial attacks,” in Proc. Advances Neural Inf. Process. Syst., 2020.
>
> [R1.4] J. Topping, F. Di Giovanni, B. P. Chamberlain, X. Dong, and M. M. Bronstein, “Understanding over-squashing and bottlenecks on graphs via curvature,” in Proc. Int. Conf. Learn. Representations, 2022.
>
> [R1.5] S. Zhu, S. Pan, C. Zhou, J. Wu, Y. Cao, and B. Wang, “Graph geometry interaction learning,” in Proc. Advances Neural Inf. Process. Syst., 2020.
>
> [R1.6] Q. Zheng, X. Zou, Y. Dong, Y. Cen, D. Yin, J. Xu, Y. Yang, and J. Tang, “Graph robustness benchmark: Benchmarking the adversarial robustness of graph machine learning,” in Proc. Advances Neural Inf. Process. Syst. Track on Datasets and Benchmarks, 2021.

---

> > ### Author Response · Authors · 2022-08-02
> > **Table R6**
> >
> > | Ratio of nodes/edges modified | Feature perturbation |     Beltrami     |     |
> > |:-----------------------------:|:--------------------:|:----------------:|:---:|
> > |         20\% / 20\%       |   $\epsilon=0.01$    | 73.73 $\pm$ 0.86 |     |
> > |         40\% / 40\%         |   $\epsilon=0.01$    | 73.28 $\pm$ 1.36 |     |
> > |         60\% / 60\%         |   $\epsilon=0.01$    | 72.46 $\pm$ 1.48 |     |
> > |         80\% / 80\%         |   $\epsilon=0.01$    | 72.31 $\pm$ 2.46 |     |
> > |         20\% / 20\%         |    $\epsilon=0.1$    | 61.56 $\pm$ 1.06 |     |
> > |         40\% / 40\%         |    $\epsilon=0.1$    | 52.15 $\pm$ 2.31 |     |
> > |         60\% / 60\%         |    $\epsilon=0.1$    | 44.03 $\pm$ 0.79 |     |
> > |         80\% / 80\%         |    $\epsilon=0.1$    | 40.30 $\pm$ 0.91 |     |
> > |         80\% / 80\%         |     $\epsilon=1$     | 39.55 $\pm$ 3.46 |     |
> > |         80\% / 80\%         |     $\epsilon=2$     | 37.09 $\pm$ 3.02 |     |
> > |         80\% / 80\%         |     $\epsilon=5$     | 37.16 $\pm$ 1.88 |     |
> > |         80\% / 80\%         |    $\epsilon=10$     | 37.16 $\pm$ 0.33 |     |
> >
> > __Table R6__: Node classification accuracy (%) on adversarial examples generated from
> > PGD under black-box *modification* setting where different number of
> > modified nodes and edges are applied. Experiments are conducted on Cora
> > dataset.

---

> ### Author Response · Authors · 2022-08-02
> **Response 3-5 (to Clarify Reviewer 1 UKSj's Weakness 3, Questions Section, and Limitations Section)**
>
>
> __Response 3.__
>
> (1). Thank you for your insightful suggestion. We have summarized the time-complexity in Table R1, which have also been included in our revision. The time is computed by averaging 500 diffusion operations. We can see that Beltrami and GRAND/BLEND have similar time complexity to the two defenders GNNGuard and GCNSVD small step sizes but incur more computation time than the other GNNs to complete a diffusion process. Heat, unlike Beltrami and GRAND/BLEND, does not use attention and has the lowest time complexity among all the neural PDEs.
>
>
>
> |  PDE solvers  |  Param.   | Beltrami | GRAND/BLEND |  Heat  |     |     |
> |:-------------:|:---------:|:--------:|:-----------:|:------:|:---:|:---:|
> | Implicit Adam | $\tau=1$  |  9.8ms   |    6.6ms    | 3.0ms  |     |     |
> |               | $\tau=2$  |  17.0ms  |   11.2ms    | 4.0ms  |     |     |
> |               | $\tau=10$ |  48.1ms  |   46.6ms    | 9.2ms  |     |     |
> | Explicit Adam | $\tau=1$  |  10.0ms  |    6.8ms    | 3.0ms  |     |     |
> |               | $\tau=2$  |  16.8ms  |   11.2ms    | 3.6ms  |     |     |
> |               | $\tau=10$ |  32.6ms  |   21.0ms    | 5.8ms  |     |     |
> |    Dopri5     |    \-     |  66.0ms  |   20.0ms    | 13.0ms |     |     |
>
>
> | RobustGCN | GNNGuard | GCNSVD |  GAT  | GraphSAGE |  GIN  | APPNP |
> |:---------:|:--------:|:------:|:-----:|:---------:|:-----:|:-----:|
> |   0.6ms   |  13.2ms  | 9.0ms  | 1.8ms |   0.8ms   | 1.0ms | 1.6ms |
>
> __Table R1__: Top: Average time spent on a Beltrami diffusion process, i.e., time to
> solve (17) and a counterpart in GRAND/BLEND, when different PDE solvers
> are applied and multiple step size options are tested for each solver.
> Bottom: Average time spent on an aggregation step using different GNNs.
> Experiments are conducted on the Citeseer dataset.
>
>
>
> (2). Thank you for pointing out the inconsistencies. We have included the missing results in Table R2. and Table R3. More specifically, we add the model performance under TDGIA attacks for all datasets that are missing in Table 4 in the main paper and Table S2 in the supplementary material. The inconsistency between Table 4 and Table 1 is caused by using different step sizes $\tau=2$ for Table 1 and $\tau=1,10$ for Table 4. Moreover, we ran 10 independent experiments for Table 1 and another 10 independent experiments for Table 4. Therefore, the statistics summarized in these two tables look slightly different. The experiment settings have also been clarified in the revision.
>
> |  PDE solvers  |  Param.   |      Clean       |      SPEIT       |      TDGIA       |
> |:-------------:|:---------:|:----------------:|:----------------:|:----------------:|
> | Implicit Adam | $\tau=1$  | 70.59 $\pm$ 2.26 | 64.64 $\pm$ 2.60 | 65.62 $\pm$ 0.96 |
> |               | $\tau=2$  | 70.14 $\pm$ 1.80 | 66.46 $\pm$ 1.33 | 65.77 $\pm$ 1.28 |
> |               | $\tau=10$ | 70.22 $\pm$ 0.70 | 64.14 $\pm$ 1.00 | 65.75 $\pm$ 1.65 |
> | Explicit Adam | $\tau=1$  | 69.72 $\pm$ 1.14 | 62.88 $\pm$ 2.70 | 64.81 $\pm$ 1.62 |
> |               | $\tau=2$  | 69.59 $\pm$ 0.92 | 65.05 $\pm$ 1.39 | 65.38 $\pm$ 1.35 |
> |               | $\tau=10$ | 69.91 $\pm$ 0.96 | 64.76 $\pm$ 1.34 | 65.52 $\pm$ 3.17 |
> |    Dopri5     |    \-     | 70.91 $\pm$ 0.98 | 66.96 $\pm$ 1.46 | 67.01 $\pm$ 1.94 |
>
> __Table R2__: Node classification accuracy (%) using graph neural PDEs induced from
> Beltrami flow, when different PDE solvers are applied. Experiments are
> conducted on Citeseer dataset.
>
> In our final version, we will make all results consistent throughout the paper.
>
> (3). Please refer to our Response 3-(1). (3). (4). to Reviewer 3 and Table R7. We explain the number of layers, Lipschitzness of the network, and over-smoothing problem in detail there.
>
>
>
>
> __Response 4.__ Regarding the Questions section:
>
> (1). The theoretical analysis in Proposition 1 in the main paper and Lemma S1 in the supplementary material show robustness against graph topology perturbation, while the theoretical analysis in Proposition 2 proves Lyapunov stability for node attribute perturbation. The experiments in the previous submitted version mainly deal with the graph topology perturbation. To further validate the model robustness against node attribute perturbation, in this rebuttal, we include modification attacks where attackers can directly flip the original graph’s edges and perturb the features of the nodes.
>
>
> However, the experiments indicate that our model only prevents small node attribute perturbations (but still strongly prevents topology perturbations) and thus has weak robustness against node attribute perturbations. We refer the reviewer to Table R6 and Response 1 to Reviewer 3 for more details about node attribute perturbation experiments.
>
> (2). We refer the reviewer to Response 1-(2) for the details.
>
> (3). We refer the reviewer to Response 3-(1) for the details.
>
>
> __Response 5.__ Regarding the Limitations section: We hope our explanations above have mitigated the reviewer’s concerns!

---

> > ### Author Response · Authors · 2022-08-02
> > **Table R3**
> >
> > |     Dataset     | Attack  |       Beltrami       |       RobustGCN        |        GNNGuard        |      GCNSVD      |          GAT           |       GraphSAGE        |          GIN           |         APPNP          |
> > |:---------------:|:-------:|:--------------------:|:----------------------:|:----------------------:|:----------------:|:----------------------:|:----------------------:|:----------------------:|:----------------------:|
> > |     Flickr      | *clean* |   49.40 $\pm$ 0.12   |    47.66 $\pm$ 0.00    |           \-           |        \-        |  **54.45 $\pm$ 0.57**  |    53.50 $\pm$ 0.02    |    53.57 $\pm$ 0.29    | ***54.08 $\pm$ 0.14*** |
> > |                 |  SPEIT  | **49.79 $\pm$ 0.68** |    6.57 $\pm$ 0.00     |           \-           |        \-        |    6.57 $\pm$ 0.00     | ***49.71 $\pm$ 0.00*** |    49.71 $\pm$ 0.00    |    6.57 $\pm$ 0.00     |
> > |                 |  TDGIA  |   49.47 $\pm$ 0.50   | ***52.95 $\pm$ 1.58*** |           \-           |        \-        |    50.38 $\pm$ 0.20    |    50.21 $\pm$ 0.11    |    50.14 $\pm$ 0.25    |  **54.28 $\pm$ 0.59**  |
> > |    Coauthor     | *clean* | **95.83 $\pm$ 0.30** |    87.75 $\pm$ 0.23    |    92.56 $\pm$ 0.16    |        \-        |    92.75 $\pm$ 0.15    | ***94.53 $\pm$ 0.21*** |    84.91 $\pm$ 0.32    |    87.67 $\pm$ 0.16    |
> > |                 |  SPEIT  | **94.83 $\pm$ 0.12** |    87.62 $\pm$ 0.29    | ***92.56 $\pm$ 0.16*** |        \-        |    2.59 $\pm$ 1.46     |   39.44 $\pm$ 13.97    |   39.44 $\pm$ 13.97    |    87.66 $\pm$ 0.16    |
> > |                 |  TDGIA  | **95.06 $\pm$ 0.21** |    87.3 $\pm$ 0.29     |           \-           |        \-        |   65.32 $\pm$ 13.04    | ***87.97 $\pm$ 3.72*** |    85.12 $\pm$ 0.33    |    87.54 $\pm$ 0.13    |
> > | Amazon Computer | *clean* |   87.86 $\pm$ 0.30   |    86.22 $\pm$ 0.54    |    88.77 $\pm$ 0.05    | 74.79 $\pm$ 0.68 | ***89.21 $\pm$ 0.60*** |  **89.92 $\pm$ 0.33**  |    86.44 $\pm$ 0.23    |    82.66 $\pm$ 1.54    |
> > |                 |  SPEIT  |   84.90 $\pm$ 0.61   |    86.33 $\pm$ 0.62    |  **88.62 $\pm$ 0.05**  | 26.79 $\pm$ 1.25 |   26.88 $\pm$ 16.68    |    29.19 $\pm$ 9.35    | ***86.44 $\pm$ 0.23*** |    82.62 $\pm$ 1.55    |
> > |                 |  TDGIA  |   85.50 $\pm$ 0.15   |  **86.69 $\pm$ 0.51**  |           \-           |        \-        |   55.45 $\pm$ 23.07    |   63.14 $\pm$ 10.59    | ***86.65 $\pm$ 0.57*** |    83.44 $\pm$ 1.59    |
> >
> > __Table R3__: Node classification accuracy (%) on adversarial examples generated by
> > SPEIT method. We denote those experiments that are computationally too
> > heavy to run by “-”. The best and the second-best result for each
> > criterion are highlighted in **bold** and ***bold and italic***
> > respectively.

---

> ### Author Response · Authors · 2022-08-08
> **To reviewer UKSj**
>
> Dear Reviewer UKSj, thank you very much again for your helpful comments and for helping us make
> our paper better! Could we kindly ask whether our point-by-point responses and the uploaded paper
> revision have addressed your main concerns about our work?

---

### Author Response · Authors · 2022-08-04
**Summary of the major changes made in the revision**

We sincerely thank all the reviewers for their helpful comments and suggestions! We now have uploaded the rebuttal version of our paper together with the  supplementary material where the revisions are marked in cyan.
We added more discussion in the main paper so that some experiments (which were in the main paper) are now placed in the supplemental material due to the space limitation. Please refer to the supplementary material for all the extensive experiments that have been conducted during the rebuttal phase.

Here is the summary of the major changes we made in the revision:

    1. We modify the title, abstract, and introduction to make our contribution and topic clearer.
    2. We include more discussion about the experimental results, and connect the empirical results with the theoretical results more closely.
    3. We provide more details and explanations about the attack settings used in our experiments in the supplemental material.
    4. We include additional extensive experiments and ablation studies in the supplemental material to have a better understanding of our models including their limitations.
    5. We test the model's robustness with different attack setups: 1) black-box injection attacks, 2) black-box modification attacks and 3) white-box injection attacks.
    6. We compute the time complexity of all neural PDEs and compare them with non-PDE GNNs.
    7. From experiment results, we shed light on some potential future works such as unveiling the effect of heat diffusivity on robustness, how to make neural PDEs more scalable and how to defend node attribute perturbation more effectively.

---

### Author Response · Authors · 2022-08-07
**Do let us know if there are any concerns that may need our clarification before discussion period ends**

Dear reviewers,

The author-reviewer discussion period will be closed in two days. If there are any concerns that may need our further clarification, please let us know before the discussion period ends.

We sincerely thank all the reviewers for their enormously valuable suggestions or comments! We also thank all the reviewers for their efforts made through the entire reviewing process!

---

### Meta-Review · Area_Chair_oj7S · 2022-08-30

**Recommendation:** Accept
**Confidence:** Certain

**Metareview:**

This paper studies the robustness properties of graph neural partial differential equations and empirically demonstrates that graph neural PDEs are intrinsically more robust against topology perturbation compared to other graph neural networks. The reviewers found the experiments extensive and convincing.  The authors provided an extensive and detailed discussion in the rebuttal phase and answered the questions raised by the reviewers. Overall, the reviewers believe that the paper discusses an interesting and important topic, but also provided some comments for improvements, such as comparing the PGD attacks with other GNNs / defense methods.

**Award:**

No

---

### Decision · Program_Chairs · 2022-09-14

Accept